# Thermostable ancestors enabled evolutionary diversification of promiscuous chemical defence enzymes

Raine E S Thomson[1], Yosephine Gumulya[1], Anthony W Bengochea (iD)[1], Gabriel Foley[1], Julian Zaugg[2], Anna Aagaard[3], Connie M Ross[1], James Beckett[1], Mikael Bodén (iD)[1], Ulrik Jurva[3], Martin A Hayes (iD)[4], Shalini Andersson[4] & Elizabeth M J Gillam (iD)[1✉]

## Abstract

Enzymes are generally believed to evolve from promiscuous ancestors to more specialized descendants under some selection pressure related to their function. However, enzymes whose function depends on substrate promiscuity have not been studied. Here, we show that a group of highly diverse, xenobiotic-metabolizing enzymes, responsible for defense against a constantly changing battery of xenobiotic chemicals, evolved from highly thermostable ancestors. Thermostability declined in parallel with the accumulation of sequence diversity through evolution. The major lineages differed in their relative diversification, with the more stable lineage leading to greater extant sequence diversity. Thermostability was associated with a trend towards better sequestration of hydrophobic residues within the core of the protein and increased exposure of polar residues in solvent-accessible parts of the structure. Resurrected ancestral forms were active towards typical substrates and exhibited ligand-binding promiscuity comparable to, or greater than, their extant descendants. This work supports the hypothesis that robust ancestors facilitate evolutionary diversification and highlights features responsible for enhancing thermostability in a protein fold.

**Keywords** Cytochrome P450; Heme-thiolate Proteins; Ancestral Sequence Reconstruction; Thermostability; Substrate Promiscuity
**Subject Category** Evolution & Ecology

## Introduction

Enzymes are generally believed to evolve from promiscuous ancestors to specialized descendants, so enzymes whose function depends on substrate promiscuity represent an unusual case for evolution. A case in point is the enzymes responsible for defending an organism against potentially toxic environmental chemicals, for example, secondary metabolites produced by plants to discourage herbivory. This "chemical defense" function is primarily mediated in vertebrates by a battery of cytochrome P450 monooxygenases (P450s) from families CYP1, CYP2, and CYP3. These xenobiotic-metabolizing P450 enzymes catalyze an extraordinarily versatile range of biotransformations on an ever-expanding range of chemical structures. Monooxygenation effects chemical defense by making a lipophilic compound more water-soluble and thereby more easily eliminated in urine or conjugated to polar metabolites such as sulfate or glucuronic acid. The resulting structural change usually also affects biological activity mediated by receptors or other targets, but not necessarily negatively. Certain subfamilies of vertebrate P450s (esp. CYP2A, CYP2B, CYP2C, CYP2D, CYP3A, and CYP1A) are believed to have diversified through evolution to metabolize xenobiotic chemicals of diverse structure, partly in response to the elaboration of secondary metabolites by plants, in a form of animal *vs.* plant evolutionary "chemical warfare". In humans, specific extant P450 forms show wide and overlapping substrate ranges that appear to be related to their ability to adopt different conformations when binding alternative substrates (Ekroos and Sjögren, 2006).

Ancestral sequence reconstruction (ASR) has allowed the inference and resurrection of ancestral proteins from numerous protein families as far back in evolution as ~2.3 billion years ago (Ga; reviewed in (Gumulya and Gillam, 2017)), and has recently been applied to P450s (Gumulya et al, 2018; Gumulya et al, 2019; Hartz et al, 2021; Foley et al, 2022; Harris et al, 2022). A common, albeit not universal, observation has been that ancestral forms of a given protein tend to be more thermostable than their extant counterparts. Maximum likelihood methods for ASR have been proposed to modestly overestimate thermostability (Williams et al, 2006). However, the increases seen (up to ~30 °C rather than the few degrees predicted by simulations) and the marked difference in

[1]School of Chemistry and Molecular Biosciences, The University of Queensland, St. Lucia, Brisbane, QLD 4072, Australia. [2]Australian Centre for Ecogenomics, School of Chemistry and Molecular Biosciences, The University of Queensland, St. Lucia, Brisbane, QLD 4072, Australia. [3]Drug Metabolism and Pharmacokinetics (DMPK), Research and Early Development, Cardiovascular, Renal and Metabolism (CVRM), BioPharmaceuticals R&D, AstraZeneca, Gothenburg, Sweden. [4]Discovery Sciences, BioPharmaceuticals R&D, AstraZeneca, Gothenburg, Sweden. ✉E-mail: e.gillam@uq.edu.au

sequence between ancestral and consensus inferences argue against these increases being wholly artifactual.

The increased thermostability in ancestors has been linked to thermophily in the case of proteins inferred from organisms that existed in the deep Pre-Cambrian past, where environmental temperatures are hypothesized to have been much higher than they are now (Knauth and Lowe, 2003; Robert and Chaussidon, 2006; Grossman and Joachimski, 2022). However, proteins inferred from more recent times, where ambient environmental conditions would have been milder, have also shown substantially higher thermostability than modern forms, notable examples being paraoxonase (Bar-Rogovsky et al, 2013) and P450s (Gumulya et al, 2018; Gumulya et al, 2019; Hartz et al, 2021; Harris et al, 2022).

One explanation for this unexpected observation, i.e., that ancestors from ancient mesophilic organisms are often thermostable, is that extant proteins have ultimately evolved from primordial forms that existed during periods of high ambient temperature, which would have required a high degree of intrinsic thermostability to function. Once ambient temperatures declined over time, genetic drift would have led to loss of thermostability in the absence of purifying selection pressure to maintain it (i.e., in all but the small proportion of organisms inhabiting thermophilic niches). However, the stochastic nature of genetic drift would mean that intermediate ancestors and extant forms from mild environments could retain varying degrees of thermostability, or indeed could have acquired stabilizing mutations via genetic drift.

Alternatively, a bias in the types of protein families studied by ASR might contribute to the frequent observation of more thermostable ancestors; highly diversified protein families are often of interest due to the range of activities they encompass, and so are more commonly the targets of ASR. For example, P450s and other enzymes involved in chemical defense are useful for biocatalysis due to the versatility in substrate and reaction specificity shown by the extant forms. By focusing on more diverse protein families, ASR projects may be inadvertently sampling more robust ancestors. A robust fold has been proposed to facilitate evolutionary diversification by buffering the effects of potentially destabilizing mutations (Bloom et al, 2006); thus, it may have been necessary for an ancestral protein to have been relatively stable to accommodate diversifying mutations without losing the ability to fold. While this contention is supported by directed evolution studies (Bloom et al, 2006), to date, the impact of ancestral thermostability has not been assessed by examining the natural evolution of a protein family.

We chose the CYP2 family of P450s to explore the relationship between ancestral robustness and evolutionary diversification. This enzyme family represents an interesting case study for protein evolution as it apparently underwent a rapid expansion following the Cambrian explosion, at least partly in response to animal *vs.* plant "chemical warfare". It encompasses hundreds of P450s responsible for metabolizing a huge array of exogenous and endogenous chemicals, including environmental toxins derived predominantly from plants, a "moving target" due to the co-evolution of plant biosynthetic pathways. Notably, the diversification of these families has occurred over the last ~500 million years (Ma), during which time ambient temperatures are understood to have varied over a range of ~30 °C, low latitude ocean temperatures cooling from a peak above ~50 °C before 500 Ma (Grossman and Joachimski, 2022).

Twelve ancestors from the clade incorporating nine subfamilies (CYP2A, CYP2B, CYP2C, CYP2E, CYP2F, CYP2G, CYP2H, CYP2S, and CYP2T) only found in tetrapods were resurrected using ASR and assessed for their thermostability and catalytic capabilities. The panel of enzymes was analyzed for sequence-structure trends that might explain cumulative changes in thermostability over time, with a view to retrospectively elucidating the design principles behind thermostability in natural proteins. In addition, thermostability is commonly assumed to be associated with rigidity, i.e., reduced flexibility in a protein fold. However, rigidity would be evolutionarily disadvantageous in proteins that have to exhibit substrate promiscuity to be functionally relevant. Therefore, we assessed the relationship between thermostability and substrate promiscuity in this enzyme family.

## Results

### Resurrection of CYP2 ancestors

ASR relies upon good-quality sequences and balanced coverage of different phylogenetic branches of the evolutionary tree under reconstruction. The process is limited by the computational power required to analyze large numbers of sequences. Given the very large size of the CYP2 phylogenetic tree, and the fact that the actinopterygian branches were relatively poorly characterized, we chose to reconstruct twelve ancestors from a subsection of the CYP2 family that evolved in tetrapods following their divergence from fish. This clade comprises genes from the CYP2B, CYP2C, CYP2E, CYP2F, CYP2G, CYP2H, CYP2S, and CYP2T subfamilies as well as several poorly characterized subfamilies from amphibia and sauropsids that have no representatives in mammals. These subfamilies represent the most recently diverging evolutionary branches of the CYP2 tree, with other notable CYP2 subfamilies (e.g., CYP2M, CYP2P, CYP2D, CYP2J, CYP2R, CYP2U, and CYP2W) having split off significantly earlier than the separation of the actinopterygii and tetrapods (Kirischian et al, 2011). The CYP2M subfamily, found only in fish, was included as an outgroup for the phylogenetic analysis. The phylogeny of 975 sequences of extant CYP2s that was used for the ancestral reconstruction is shown in Fig. 1. (A more detailed version of this tree showing taxonomic identifiers and bootstrap values can be seen in Fig. EV1.) The general arrangement of CYP2 subfamilies is in agreement with the phylogeny of an earlier analysis of CYP2 evolution carried out by Kirischian et al (Kirischian et al, 2011). P450s included in this 2011 study were limited to those from fully sequenced genomes, with only mammalian representatives of subfamilies CYP2A, 2B, 2C, 2E, 2F, 2G, 2S, avian representatives of subfamily CYP2H, and no reptilian sequences. Since that 2011 study, sequence databases have been significantly enriched, and we were able to retrieve representatives of these subfamilies from other tetrapod classes, including: CYP2Cs from birds and reptiles; CYP2Gs from reptiles; and CYP2Fs and CYP2Hs from reptiles. To ensure a broad sampling of sequence space in the present study, sequences were included which clustered with and encoded plausible sub/family members when compared with validated sequences of the subfamilies under consideration. Twelve internal nodes of the tree were chosen for resurrection and named according to the subfamilies to which they gave rise; e.g., CYP2ABGS gave rise to

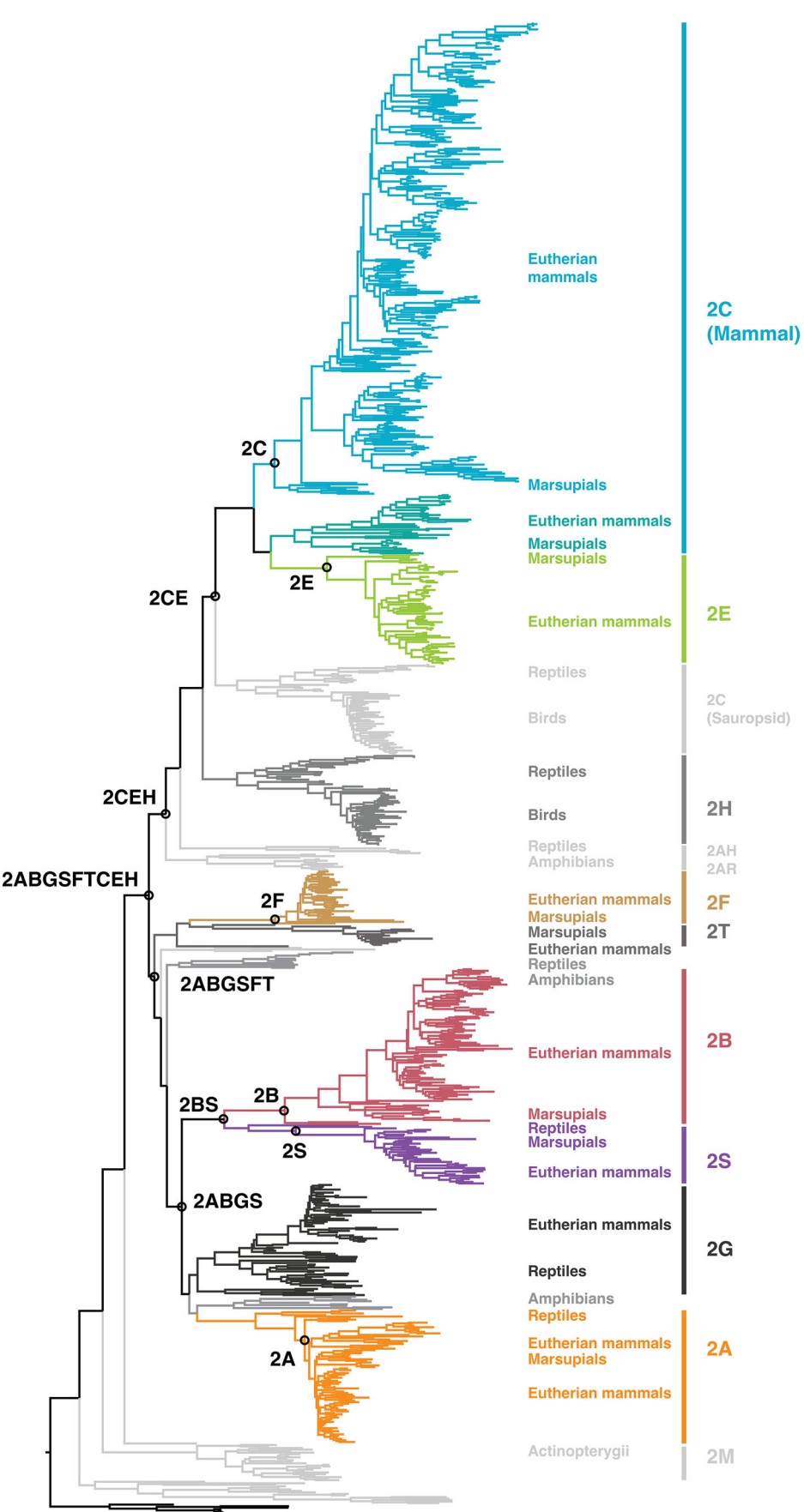

◀ **Figure 1.   CYP2 phylogeny.**

The CYP2 clade used for reconstruction incorporates the tetrapod subfamilies, 2A, 2B, 2C, 2E, 2F, 2G, 2H, 2S and 2T, and was inferred from 975 extant CYP2 sequences. Sequences were collected from NCBI and Uniprot databases using a BLAST-search of sequences with >40% similarity to characterized CYP2 forms. The CYP2M subfamily was used as an outgroup. Nodes representing the ancestors chosen for reconstruction are numbered. Ancestors are named according to the subfamilies that they gave rise to: **N851**, 2A; **N625**, 2B; **N10**, 2C; **N357**, 2E; **N562**, 2F; **N727**, 2S; **N624**, 2BS; **N623**, 2ABGS; **N8**, 2CE; **N5**, 2CEH; **N559**, 2ABGSFT; **N4**, 2ABGSFTCEH. Labeled extant forms indicate those used for comparison in all thermostability and catalytic analyses. Source data are available online for this figure.

the CYP2A, CYP2B, CYP2G, and CYP2S subfamilies. The nodes were selected to cover broad evolutionary distances as well as to represent the major xenobiotic-metabolizing lineages. The resultant selection included six younger ancestors that gave rise to single subfamilies (CYP2A, CYP2B, CYP2C, CYP2E, CYP2F, and CYP2S), four ancestors that gave rise to two, three or four subfamilies (CYP2CE, CYP2CEH, CYP2BS, CYP2ABGS) and two older ancestors that gave rise to six and nine subfamilies, respectively (CYP2ABGSFT and CYP2ABGSFTCEH). The shared amino acid identity between directly related ancestors and extant forms ranged from 51–98% (Appendix Table S1).

The ancestors from the joint reconstruction characterized here were ~98% or more identical at the amino acid sequence level to those inferred from a marginal reconstruction based on the same tree and alignment with one exception (at 96.36% identical) (Appendix Table S2). Most discrepancies involved conservative substitutions (Appendix Table S2). The joint and marginal ancestors at each node were closer to each other than would be expected from random sampling from the marginal distribution.

When expressed in *E. coli*, all reconstructed ancestors were found to produce folded proteins capable of incorporating a heme prosthetic group, evidenced by the presence of a peak at 450 nm in the Fe(II).CO *vs*. Fe(II) difference spectrum. Expression levels were generally higher for the ancestors than for the extant forms expressed in the same system (Appendix Table S3), a factor that may be due in part to optimization of the ancestral sequences for *E. coli* codon preferences or the N-terminal modifications introduced to facilitate recombinant expression in bacteria. However, expression yields also varied roughly tenfold among the ancestors, ranging from ~160 to 2000 nmol/L, a range that was comparable to yields observed previously for ancestors resurrected from the CYP3 (Gumulya et al, 2018) and CYP1 families (Harris et al, 2022). There was no clear relationship between expression levels and evolutionary distance; however, the highest expressing forms were the younger ancestors, CYP2A, CYP2E, and CYP2S, whereas the two oldest ancestors, CYP2ABGSFT and CYP2ABGSFTCEH, were amongst the least well expressed.

## Thermostability of ancestral and extant CYP2 forms

The 12 ancestors and 12 of their extant descendants were assessed for thermostability by monitoring the disappearance of the characteristic Soret peak in the Fe(II).CO *vs*. Fe(II) difference spectrum (Fig. EV2A) after heating at various temperatures for 10 or 60 min (Fig. EV2B). The temperatures at which 50% of the protein was denatured over these time periods ($^{10}T_{50}$ and $^{60}T_{50}$, respectively) are summarized in Appendix Table S3 and the $^{60}T_{50}$ measurements are correlated with evolutionary distance from the oldest ancestor in Fig. 2. Branch lengths are used to indicate the evolutionary distance between different ancestors but cannot be

used as a surrogate for historical time due to the likelihood that amino acid substitution rates vary across time, species and position in the protein structure. All ancestors were found to be more stable than their direct extant descendants ($P < 0.006$, two-tailed heteroscedastic Student's *t* test), and a general trend of increasing stability going back to the oldest ancestors was observed. The $T_{50}$ values of younger ancestors, in particular CYP2A, CYP2B, CYP2E, CYP2F, and CYP2S, showed increases of ~5–10 °C compared with the extant forms, whereas the older ancestors, namely CYP2ABGS, CYP2CE, CYP2CEH, CYP2ABGSFT, and CYP2ABGSFTCEH, displayed $T_{50}$ values that were 30–40 °C greater than the extant forms. In general, the ancestors reconstructed along the CYP2C/E/H evolutionary branch were 5–15 °C more thermostable than the corresponding ancestors reconstructed along the CYP2A/B/F/G/S/T branch, with CYP2CE and CYP2CEH showing the highest stability of all forms.

To determine whether the profound thermostability seen here was an artifact of the assay method used, the unfolding of CYP2CE was also analyzed by circular dichroism. The melting temperature ($T_m$) was found to be 76.5 °C (Appendix Fig. S1), consistent with the estimate of 75–78 °C for the $^{10}T_{50}$ and $^{60}T_{50}$ values obtained by the spectral method, indicating that the thermostability seen here was not an artifact of the analytical technique used.

## Kyte–Doolittle hydrophobicity analysis

Seeing incremental reductions in thermostability over time led us to hypothesize that the underlying molecular forces responsible for this trend were additive, such as hydrophobic/hydrophilic interactions. To assess whether thermostability was correlated with hydrophobicity, the sequences of all CYP2 forms for which thermostability was measured were quantified using the Kyte–Doolittle hydrophobicity index (Kyte and Doolittle, 1982). When calculating hydrophobicity, the solvent accessibility of each residue was taken into consideration since the stabilizing effect of any given residue is largely dependent on its solvent exposure. This analysis revealed that the most thermostable forms had a higher degree of hydrophobicity in their buried residues and a higher degree of hydrophilicity in their exposed residues when compared to the less stable forms (Fig. 3A). This trend was most apparent when looking at the whole protein (Fig. 3B). When the fold was divided into discrete structural regions the trend weakened, suggesting a cumulative effect across the entire sequence (Appendix Fig. S2). We performed a phylogenetic independent contrast analysis (Garland et al, 1992) on the 12 extant sequences to identify any correlation between the buried/exposed values and thermostability, while accounting for the phylogenetic structure. Buried residues showed a moderate correlation but without reaching statistical significance ($P$ value > 0.1) (Appendix Table S4).

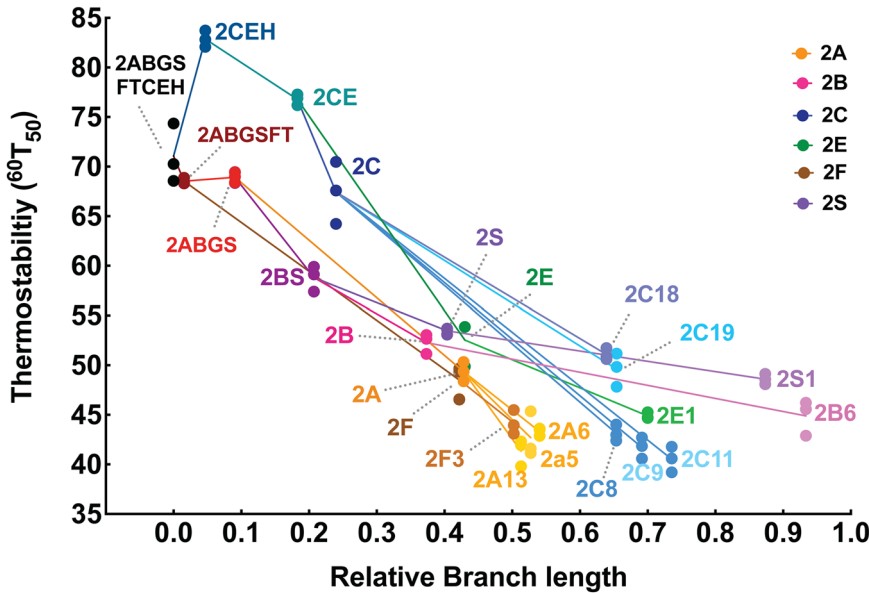

**Figure 2. Correlation between P450 thermostability and branch length in the CYP2 family.**

Relative branch length from the oldest ancestor was plotted against the $^{60}T_{50}$ (Appendix Table S3) of each form. Colors correspond to subfamily lineages (CYP2A, orange; CYP2B, pink; CYP2C, blue; CYP2E, green; CYP2F, brown; CYP2S, purple) and lines connecting points represent direct evolutionary relationships. $^{60}T_{50}$ determinations from three independent biological replicates are shown. A general trend was observed towards increased thermostability correlating with greater evolutionary distance from the extant forms. All ancestors showed significantly different $^{60}T_{50}$ values to their immediate ancestor along the same lineage, except CYP2ABGSFT (compared to CYP2ABGSFTCEH) and CYP2ABGS (compared to CYP2ABGSFT); two-tailed, heteroscedastic Student's $t$ test. All extant enzymes showed $^{60}T_{50}$ values that were significantly lower than all their cognate ancestors ($P < 0.006$, two-tailed heteroscedastic Student's $t$ test). Source data are available online for this figure.

Residue interaction networks were also analyzed among the ancestors, focusing on the greatest differences in thermostability within the least change in sequence, namely the three oldest ancestors CYP2ABGSFTCEH, CYP2CEH, and CYP2ABGSFT (95–98% identical; Appendix Table S1). CYP2ABGSFTCEH had 24 fewer interactions than its more thermostable descendant, CYP2CEH, but 14 more than CYP2ABGSFT, which was slightly but not significantly less stable (Appendix Table S3). CYP2CEH had 10 more interactions than CYP2ABGSFT. The great majority of interactions were conserved or conservatively substituted among at least two of the three ancestors, but CYP2CEH had 156, CYP2ABGSFTCEH had 115, and CYP2ABGSFT had 96 unique interactions, in line with their order of thermostability. No clear candidates could be identified as critical for gain or loss of thermostability, with changes distributed throughout the length of the sequences.

## Catalytic activity of ancestral forms towards potential endogenous substrates

Steroid hormones are a prominent group of endogenous P450 substrates, and testosterone has been reported as a substrate for most characterized CYP2s. As such, all ancestors were assessed for their ability to metabolize testosterone. Most ancestors were found to hydroxylate testosterone to form a variety of metabolites (Fig. 4; Appendix Fig. S3). The exceptions to this were CYP2S and CYP2BS, which showed no activity towards testosterone; however, notably, the closely related extant CYP2S1 is one of the few human CYP2 forms that does not apparently metabolize testosterone. The most abundant metabolites produced were 16α-, 16β-, and 2α/β-

hydroxytestosterone and androstenedione, followed by 6α-, 6β-, and 15α-hydroxytestosterone, identified based on the retention time of known standards. There was no obvious correlation between evolutionary distance from the extant forms and total testosterone turnover, however the extant forms were generally more active towards this substrate than the ancestors. Extant forms CYP2a5 and CYP2C11 showed the highest turnover, and the immediate ancestors of these forms, i.e., CYP2A and CYP2C as well as CYP2CE and CYP2CEH, were the most active of the ancestors.

## Catalytic activity of ancestral forms towards marker substrates

To further characterize their catalytic capacity, the reconstructed ancestors were assessed for activity towards various marker substrates, including alkoxyresorufins, luciferin derivatives (P450-Glo™ substrates), and coumarin. Coumarin is predominantly a substrate of forms from the CYP2A subfamily, and the CYP2A ancestor was the only ancestral form to show activity towards this substrate as assessed via the detection of umbelliferone production (Appendix Fig. S4). Across the other two classes of substrates (luciferin derivatives and alkoxyresorufins), all ancestors showed activity towards at least one compound (Fig. 5; Appendix Figs. S5 and S6). Forms CYP2E and CYP2S showed among the highest activity of the ancestors overall towards the luciferin derivatives, a contrast to their poor turnover of testosterone. Conversely, the most extensive metabolisers of testosterone among the ancestors, CYP2A and CYP2C, showed only mediocre if any activity towards the luciferin and alkoxyresorufin derivatives,

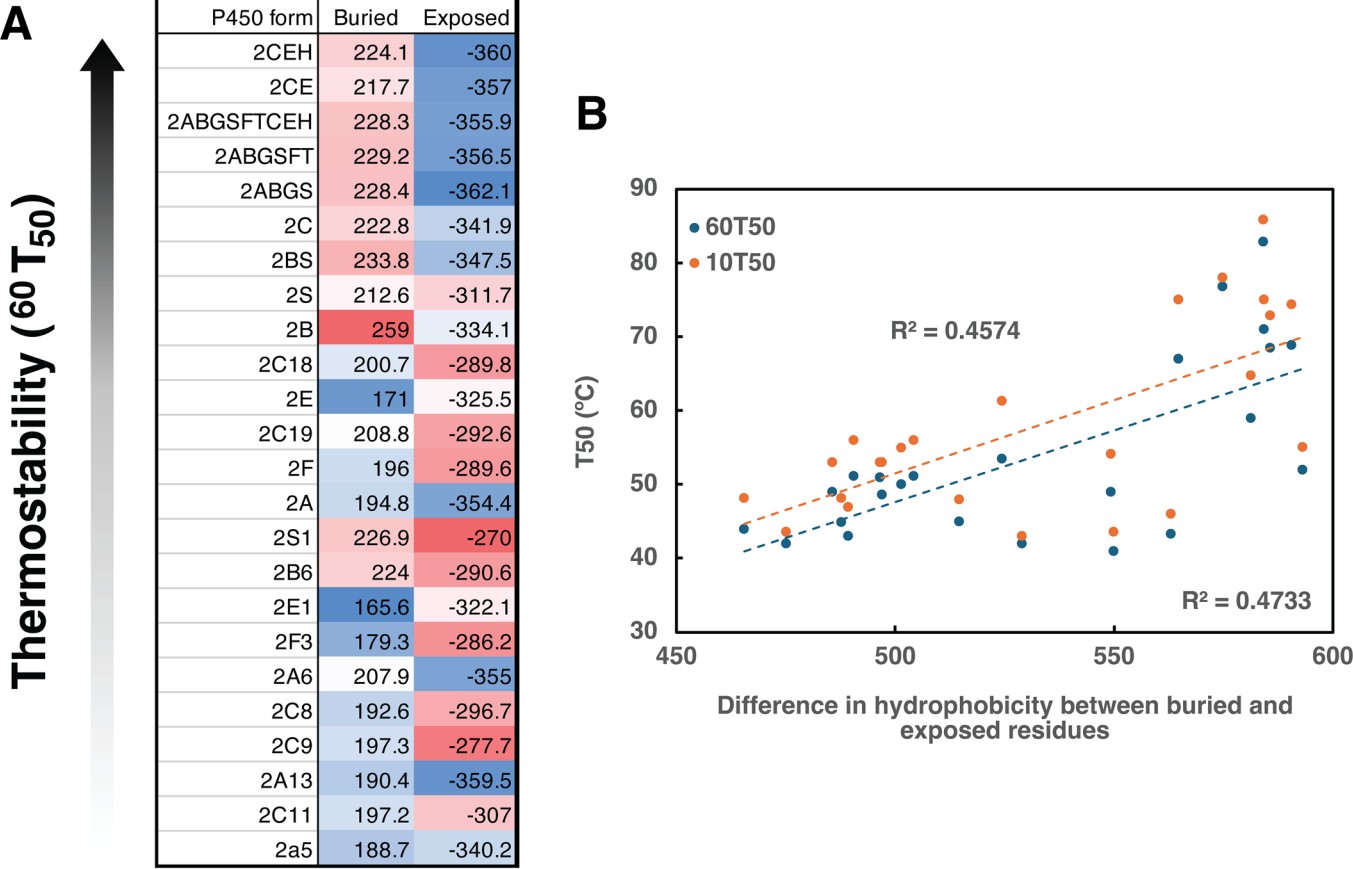

**Figure 3. Kyte–Doolittle hydrophobicity of buried and exposed residues.**

The hydrophobicity of all twelve ancestors and twelve extant forms (listed) was calculated according to the Kyte–Doolittle hydrophobicity index. Colors represent relative hydrophobicity, from most hydrophobic (red) to most hydrophilic (blue). All residues were classified as either buried or exposed according to the degree of solvent exposure of the corresponding residue in crystal structures of CYP2A6, CYP2C9, and CYP2B6. (A) The score given for each form reflects the summed Kyte–Doolittle hydrophobicity scores for all residues that are ≤30% solvent exposed (buried) or >30% solvent exposed (exposed). The heatmap refers to the scale within each column and is scaled from 0 to 100% of the range of hydrophobicity. (B) Difference in summed scores for buried minus solvent-exposed residues *vs.* thermostability. Source data are available online for this figure.

although some of their descendants, specifically cyp2a5, CYP2A13, and CYP2C11 showed some of the highest activity overall towards substrates in both groups.

Another common activity of many CYP2 forms is the hydroxylation of indole to form indigo (Gillam et al, 1999), a reaction carried out by extant P450s, cyp2a5, CYP2A6, CYP2A13, CYP2C19, CYP2E1, and CYP2F3. Indole is an endogenous compound produced by enterobacteria from the breakdown of tryptophan, and this reaction can occur during recombinant P450 expression in *E. coli*. Cultures co-expressing ancestral forms CYP2A, CYP2E, CYP2F, and CYP2CEH were found to produce a blue pigment (Appendix Fig. S7), an indication that these forms can oxidize indole to indoxyl, which then dimerizes to indigo.

Collectively, the patterns of activity of ancestors were similar to those seen for the extant forms, i.e., ancestors showed discrete but overlapping ranges of activity towards xenobiotic substrates.

## Ligand promiscuity of ancestors

Most P450s are promiscuous in their capacity to bind structurally and chemically diverse compounds due in part to a degree of

flexibility in the active site allowing it to expand and contract. It was of interest to assess what trade-off (if any) there was between stabilizing the P450 fold and maintaining ligand promiscuity. A set of 48 chemically diverse pharmaceutical compounds known to be ligands of extant P450s was screened for binding to each ancestor using a competition assay (Appendix Table S5 and Appendix Fig. S8). In general, the activity of each form was affected to some degree by many of the compounds screened. In most cases, the compounds acted as inhibitors; however, in the case of CYP2C, and other forms to a lesser extent, there were multiple compounds that enhanced activity, suggesting cooperative effects on substrate binding and/or turnover, as reported previously for other P450s (Ueng et al, 1995; Hutzler and Tracy, 2002). To assess the correlation between ligand promiscuity and stability, the diversity of the ligand set for each ancestor was quantified using the Tanimoto distance measure (Tanimoto, 1958) and the effector promiscuity was calculated as described previously (Nath and Atkins, 2008; Foti et al, 2011) except that both inhibition and activation of activity was considered and the relative change from the control (no effector) was used as a surrogate measure of ligand interaction. Based on the average Tanimoto score (Fig. EV3A,B),

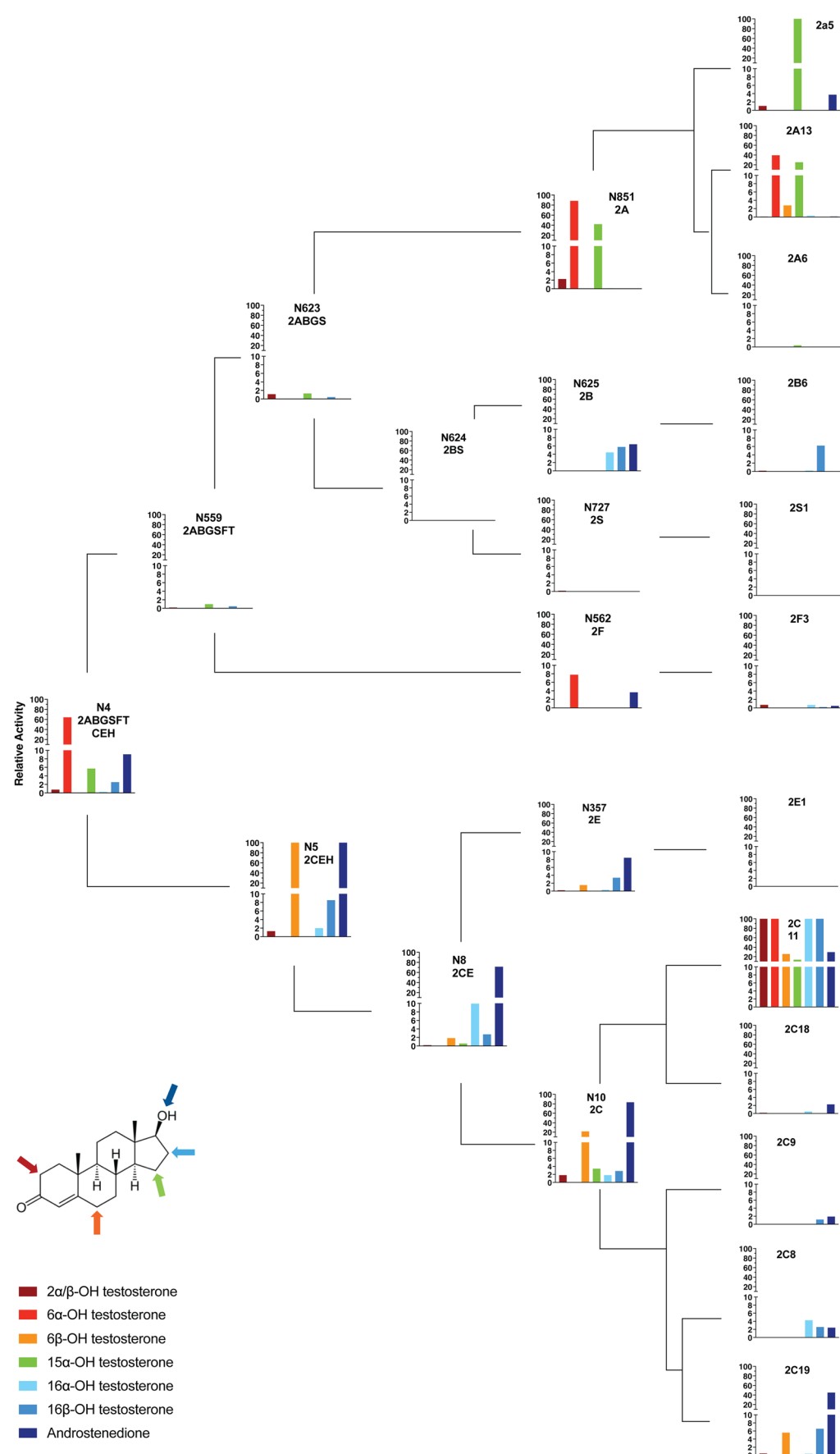

**Figure 4.  Metabolism of testosterone by ancestral and extant CYP2 forms.**

Plots illustrate the relative activity of ancestral and extant forms towards the metabolism of testosterone, normalized to the activity shown towards the substrate and metabolite in question by the most active form. The upper half of the figure shows the CYP2ABGSFT lineage, whereas the lower half shows the CYP2CEH lineage. The background signal as indicated by the hCPR-only negative controls (Appendix Fig. S3) was subtracted before calculating relative activity. Only data that are significantly greater (two-sided Student's *t* test; $P < 0.05$) than the negative controls are shown, and data represent the means of $n = 3$ technical replicates. For clarity, individual data points and statistical variation are not shown but can be seen in Appendix Fig. S3. Plots are superimposed on the evolutionary tree of the CYP2 family to show evolutionary relationships; however, branch lengths are not to scale. Activity was assessed in reactions carried out for 60 min at 37 °C in 250 µl total volume using 100 µM testosterone and 0.5 µM P450 added in bacterial membranes. Testosterone was metabolized to seven primary metabolites, which were quantified using a standard curve prepared with authentic standards. Source data are available online for this figure.

there did not appear to be any correlation between average Tanimoto distance and relative branch length or any inherent trade-off between thermostability and ligand diversity. However, the younger ancestors typically interacted with fewer ligands overall (albeit similarly diverse ligands). All ancestors showed J values close to one indicating high promiscuity and there was no discernible trend towards greater or less promiscuity through evolution of the two major lineages (Fig. EV3C,D). Spectral binding analysis showed that the relative affinity of ancestors for five typical ligands was in the high nanomolar to micromolar range typical of extant forms (Appendix Table S6).

## Discussion

The proteins resurrected here represent the most probable ancestors given the available sequence information, evolutionary model, and algorithms chosen for ancestral inference. Importantly, they do not necessarily reflect any one historically accurate ancestral enzyme but rather represent the most likely candidate of a set of possible sequences. As such, there is a degree of uncertainty in any observed structural and functional features as with any ancestral sequence reconstruction. Joint reconstruction was selected here, since the aim was to compare ancestors across a consistent reconstruction, and the GRASP tool (Foley et al, 2022) was chosen to maximize the amount of sequence information that could be used. A comparison of the ancestors predicted by joint and marginal maximum likelihood approaches showed little variation (generally less than ~2%) in the sequence of the inferred ancestors (Appendix Table S2). Likewise, the variation in sequences caused by accommodating variation in substitution rates was <2% (Appendix Table S7). The greatest effect observed on ancestor sequences was seen with the choice of tree (Appendix Table S7), as found by others (Muñiz-Trejo et al, 2025). The choice of tree was informed by the expected species relationships and a previously published phylogenetic analysis (Kirischian et al, 2011), and statistical support was low for some branch points. However, when repeated with a different tree generated in IQ-TREE2 that ignored established phylogenetic relationships, the bootstrap values were still low at some nodes. The ancestors at comparable nodes from different trees were generally > 90% identical (all were at least 85% identical; Appendix Table S7), and the positions that were inferred differently corresponded to those showing lower posterior probability in the marginal reconstruction. Importantly, the tetrapod CYP2 sequence clade selected for reconstruction is relatively rich in sequence information from heavily studied higher vertebrates, with representation across different animal classes and

at key evolutionary branchpoints, all of which are conducive to greater accuracy in the ancestral inference (Foley et al, 2022), since they provide more information on which to build a more accurate tree.

Resurrected ancient proteins have been reported to demonstrate superior thermostability compared to their extant descendants in previous ASR studies (Gaucher et al, 2008; Perez-Jimenez et al, 2011; Akanuma et al, 2013; Butzin et al, 2013; Risso et al, 2013; Akanuma et al, 2015; Whitfield et al, 2015; Trudeau et al, 2016; Garcia et al, 2017), but many of these earlier reconstructions have focused on ancient bacterial and archaeal protein families that originated in Pre-Cambrian conditions, where thermostability was a necessary characteristic for functionality, due to the hot marine conditions that were likely experienced by early life (Robert and Chaussidon, 2006; Garcia et al, 2017). Logically, proteins from mesophilic vertebrates emerging during or after the Ordovician period (~450 Ma), such as the tetrapod P450 ancestors described here, should not have required high stability, as these organisms would have lived in milder ambient temperatures. However, recent studies have shown that enhanced stability may be a general trait of ancestral P450s from vertebrate animals: progenitors of the CYP3 and CYP1 families and CYP2D and CYP11A subfamilies, all of which would have been present in ancient vertebrates, displayed markedly higher thermostability compared to their extant counterparts (Gumulya et al, 2018; Gumulya et al, 2019; Hartz et al, 2021; Harris et al, 2022). A similar increase in thermostability has also been reported for ancestors of another xenobiotic-metabolizing enzyme (Trudeau et al, 2016). Thus, we hypothesized that the thermostability observed in P450s reconstructed from ~400 to 450 Ma may have been an evolutionary relic of more ancient forms that would have needed to tolerate much hotter ambient temperatures. We sought to explore the change in thermostability across evolutionary time using a clade of the highly diverse, xenobiotic-metabolizing CYP2 family as a model system.

Twelve ancestors representing various evolutionary stages of the tetrapod clade of the CYP2 family were inferred and resurrected (Fig. EV4). All ancestors incorporated a heme prosthetic group, producing a typical P450 Soret peak indicating an intact heme-thiolate linkage. The loss of the characteristic heme-thiolate spectrum after heat treatment was used to assess their thermo-stability in comparison to twelve of their extant descendants. All ancestors were found to be significantly more stable than their directly related extant forms ($P < 0.006$, two-tailed heteroscedastic Student's *t* test), and a general trend of increasing thermostability with evolutionary distance from the extant forms was observed; the most stable ancestors showed $T_{50}$ values of up to ~40 °C higher than those of their extant counterparts.

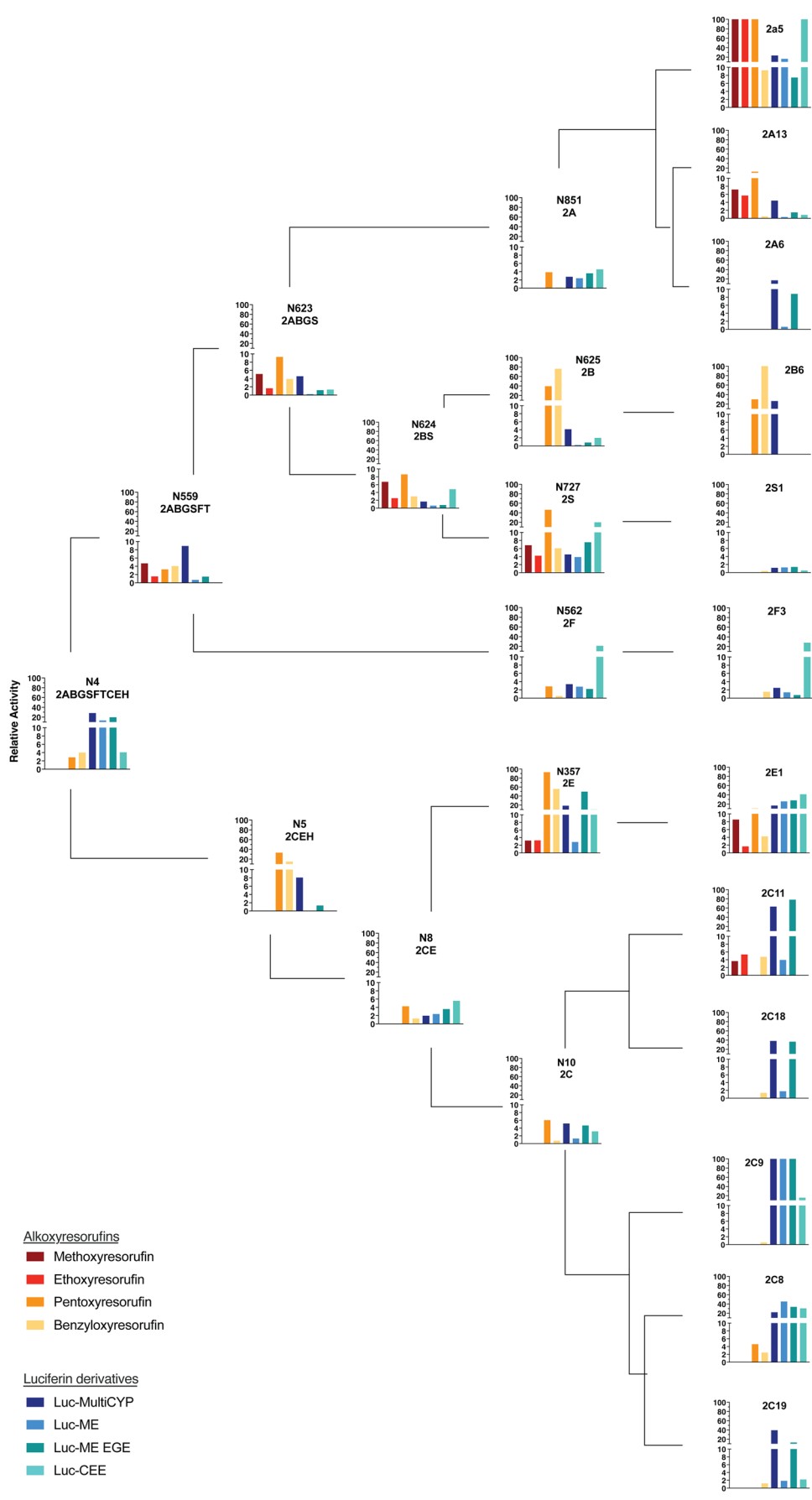

**Figure 5.    Relative activity of CYP2 ancestors towards 7-alkoxyresorufin and luciferin derivatives as marker substrates of extant human CYP2 forms.**

Plots illustrate the relative activity of ancestral and extant forms towards the dealkylation of eight CYP2 probe substrates: 7-ethoxyresorufin, 7-methoxyresorufin, 7-benzyloxyresorufin, 7-pentoxyresorufin, luciferin-MultiCYP, luciferin-ME, luciferin-ME EGE, and luciferin-CEE, normalized to the maximal activity shown towards that substrate by any form. The upper part of the figure shows the CYP2ABGSFT lineage, whereas the lower part shows the CYP2CEH lineage. The background signal as indicated by the negative controls (hCPR-only in the case of the 7-alkoxyresorufin assay; Appendix Fig. S5) or without NADPH in the case of the luciferin assay (Appendix Fig. S6) was subtracted before calculating relative activity. Only data that are significantly greater (two-sided Student's $t$ test; $P < 0.05$) than negative controls are shown shown and columns represent the means of $n = 3$ technical replicates. For clarity, individual data points and statistical variation are not shown but can be seen in Appendix Figs. S5 and S6. Plots are superimposed on the evolutionary tree of the CYP2 family to show evolutionary relationships; however, branch lengths are not to scale. All reactions were carried out at 37 °C. Alkoxyresorufin O-dealkylation assays were performed using bacterial membranes normalized to a P450 concentration of 25 nM, and a substrate concentration of 5 μM in a total volume of 100 μl. Luciferin-based assays were performed with bacterial membranes, using a P450 concentration of 40 nM and a substrate concentration of 50 μM for luciferin-MultiCYP, 100 μM for luciferin-ME, and 30 μM for luciferin-ME EGE and luciferin-CEE in a volume of 25 μl. Source data are available online for this figure.

Along the CYP2ABGSFT lineage, the three most basal ancestors displayed similar stability with $^{60}T_{50}$s of ~70 °C. Thereafter, there appeared to have been a continuous, steady loss of thermostability with each successive ancestor in this lineage. However, the divergence of the other major lineage, CYP2CEH, appeared to include one or more mutational events that resulted in a ~10 °C increase in $^{60}T_{50}$ of CYP2CEH compared to the oldest ancestor, CYP2ABGSFTCEH, from which it differs in 48 of 472 residues. This highlights that although most mutations, both functionally neutral and those endowing new enzymatic functions, are likely to have a destabilizing effect (Guerois et al, 2002; Tokuriki et al, 2008), random genetic drift may also lead to improvements in stability (Bershtein et al, 2008). Following this putative stabilization event, this lineage also shows a steady loss of stability over time. The younger ancestors of single subfamilies, CYP2A, CYP2B, CYP2E, CYP2F, and CYP2S are all likely to be of similar evolutionary age since they all existed in the common ancestor of the mammals. Despite arising from diverging branches, all display relatively similar stabilities, varying <4 °C in $T_{50}$ values. Although these younger forms are ~10–30 °C less stable than the older ancestors, they are all ~5–10 °C more stable than their direct descendants ($P < 0.035$, two-tailed, heteroscedastic Student's $t$ test).

These trends are consistent with thermostability being an evolutionary relic from enzymes that existed in very hot environments in primordial times, that was lost through genetic drift in the absence of selection pressure to maintain it. Since most mutations are deleterious (Tokuriki et al, 2008), genetic drift would be expected to lead to a loss of thermostability with the accumulation of mutations, until a level was reached where enzyme function began to be impaired. P450s are found in all domains of life, so could have been present in the last universal common ancestor (LUCA). However, they appear to have been transferred horizontally to archaea (Ngcobo et al, 2023), so they probably first appeared in an ancient bacterium around the time of the Great Oxidation Event ~ 2.4 billion years ago (Ga). Estimates put the temperature of the Earth in the Archean era at 55–85 °C with a decline thereafter. Temperatures around 2–2.4 Ga are estimated to have been in excess of ~60 °C (Robert and Chaussidon, 2006). The CYP2 clade studied here dates from the earliest tetrapods, i.e., ~400–420 Ma (George and Blieck, 2011), meaning that the oldest ancestor resurrected here corresponds to vertebrates that may have inhabited marine environments with average global temperatures of ~15–25 °C (compared to

~15 °C today) (De Vleeschouwer et al, 2024) with temperatures up to ~40 °C in low latitudes (Grossman and Joachimski, 2022). A very recent analysis of average global temperatures during the Phanaerozoic Era (Judd et al, 2024) suggests that over the last 485 million years, the global mean temperature has varied between 11 and 36 °C but 22 and 42 °C in the tropics. The peak hothouse temperature is proposed to have occurred ~89–94 Ma and the most recent ~55.8 Ma during the Paleocene–Eocene Thermal Maximum, with a steady decline in temperature thereafter.

It is impossible to accurately evaluate whether such temperatures would have imposed any significant selection pressure on the maintenance of thermostability in proteins in putative ancestral vertebrates without knowing which latitudes they inhabited, and their ability to regulate their temperature by either physiological (homeothermic) or behavioral means (e.g., moving to cooler environments). Nor is it possible to define the time at which a particular putative ancestor would have existed except within very broad windows of evolutionary time. For example, the CYP2C and CYP2E subfamilies are both present in marsupials and eutherian mammals, so logically must have been in their most recent common ancestor (MRCA). They both derive from the CYP2CE ancestor which covers (only) the same clades. Logically, the CYP2CE must have predated the CYP2C and CYP2E ancestors, but we have no way of ascertaining when the duplication occurred. It must have occurred between the MRCA of the eutherian mammals and marsupials at ~160 MYA and the MRCA of the mammals and sauropsids (reptiles and birds) at ~319 MYA, but when exactly—or even approximately—in this ~159 MY period is unclear.

Bearing in mind these caveats, it is notable that the average melting temperature of the proteome was ~10–20 °C higher than the respective optimal growth temperature (OGT) across a range of organisms exhibiting different OGTs (Jarzab et al, 2020). The distribution of melting temperatures ranged ~10–20 °C above and below the average for the three vertebrates analyzed. Thus, average melting temperatures of 50–60 °C would not be unreasonable for proteins in organisms surviving hothouse environments of ~30–40 °C, and the range could extend up to 80 °C. We cannot assign precise ages to any of the hypothetical ancestral proteins resurrected here, but the deeper ancestors (i.e., CYP2ABGSFTCEH, CYP2ABGSFT, CYP2ABGS, CYP2CEH, CYP2CE, and CYP2BS, that could have existed closer to times of hothouse temperatures than the single subfamily ancestors) showed $^{60}T_{50}$ values from ~59

to 83 °C, placing them toward the top of, or above, the expected range.

The CYP2 family is the largest and most diverse of the P450 families in vertebrates, with 33 subfamilies across vertebrate species (Nelson, 2003). The exceptional stability of the basal CYP2 forms may have permitted this extensive degree of diversification, which is consistent with the idea that high stability contributes to the evolvability of a protein (Bloom et al, 2006; Besenmatter et al, 2007) by allowing destabilizing but functionally useful mutations to be tolerated without compromising the protein fold. In particular, the CYP2CEH lineage shows the greatest diversification within the clades of CYP2 forms studied to date (CYP2D, CYP2U, CYP2CEH, and CYP2ABGSFT), and the CYP2CEH ancestor was also the most stable ancestor of those resurrected to date (Gumulya et al, 2018; Gumulya et al, 2019; Foley et al, 2022). Moreover, the CYP2C ancestor has $T_{50}$ values that are significantly (13.5–22 °C) higher than the other single subfamily ancestors ($P < 0.02$, two-tailed, heteroscedastic Student's $t$ test). This extra stability may be the reason the CYP2C lineage has given rise to roughly four times the number of functional extant forms than seen in other CYP2 subfamilies (Fig. 1). The high stability of the CYP2C ancestor is likely to be a result of the stabilizing mutations introduced into the CYP2CEH form.

Evolutionary simulations have suggested that proteins evolve quickly to a point where they are marginally stable (Goldstein, 2011), which is consistent with the idea that thermostability is lost by genetic drift, but would imply a close temporal relationship should be seen between global temperature shifts and the changes in thermostability observed in proteins (assuming generation times are short relative to the timescale of climate changes). It is not clear how the assumptions used for these simulations relate to the natural evolution of a protein family like the CYP2s.

In contrast, prospective in vitro evolution studies on a bacterial P450 have revealed that neutral evolution leads to excess mutational robustness via enhancement of protein stability in populations of a sufficient size and diversity (Bloom et al, 2007). While this effect would be expected to be more significant in microbial populations than vertebrates (Bloom et al, 2007), given the scope of the CYP2 family under analysis here, and the degree of polymorphism evident in extant enzymes (Gaedigk et al, 2021), the effective population of P450s under consideration here may be sufficiently large and diverse to fit these criteria (or may have been at some point in the past), leading to elevated protein stability and enhanced tolerance of mutations, explaining in part the maintenance of thermostability at mesophilic temperatures.

Importantly, thermostability not only confers resistance to elevated temperature (Trudeau et al, 2016). The functional role of most P450s in the tetrapod CYP2 clade analyzed here is dependent upon the ability to evolve in response to animal vs. plant chemical warfare, so there may be selection pressure to maintain stability and thereby mutational robustness. It is straightforward to imagine bottlenecks where having a high degree of polymorphism in an enzyme used for detoxification (i.e., the ability to rapidly evolve function in a chemical defense gene) might have conferred a survival advantage. Selection pressure from dietary toxins may be mild under most conditions but potentially very strong, e.g., after a famine where foraging is restricted to specific plants containing toxic secondary metabolites.

We cannot exclude the possibility that the enhanced thermostability of the P450 ancestors is due to a "consensus effect" associated with the use of maximum likelihood methods, as proposed by Williams et al (Williams et al, 2006). However, it is useful to note the extensive differences between the consensus and ancestral proteins (Appendix Table S8); for example, the CYP2C ancestor and the corresponding consensus protein differed in 80 residues (82.7% amino acid sequence identity), whereas the oldest ancestor differed from its respective consensus in 112 residues (75.8% amino acid sequence identity). Indeed, the most closely related ancestral and consensus proteins were also the least thermostable (i.e., the youngest ancestors). Such an effect would be expected to lead to an increase in $T_{50}$ value of ~5 °C based on the estimated artifact (Williams et al, 2006; Thomson et al, 2022), much less than the increases seen here. In addition, stabilization is not seen consistently in ancestors derived by ML approaches: a recent study of a bacterial P450 phylogeny failed to show a systematic increase in thermostability in ancestral forms reconstructed using the same methodology. Rather, the stability instead mirrored the likely physiological niche of the enzymes (Jones et al, 2024).

In studies comparing thermophilic vs. mesophilic proteins and where proteins have been engineered for stability, hydrophobic core packing has been shown to affect thermostability, along with increases in salt bridges, disulfide bonds, and stacking of aromatic residues. No significant differences were seen in salt bridges or aromatic stacking in the more vs. the less thermostable ancestors, and P450s are generally free of disulfide bonds, but it was hypothesized that additive hydrophobic interactions may be responsible for the differences in thermostability between ancestors. Although the overall number of hydrophobic residues was not found to be significantly different between more and less stable forms, the context of the residues, particularly whether they are buried or exposed, may ultimately determine their stabilizing effect. When residues were grouped as either "buried" or "exposed" based on solvent accessibility there was a trend towards buried residues being more hydrophobic and exposed residues more hydrophilic in the more stable forms (Fig. 3), but the difference failed to reach statistical significance (Appendix Table S4).

It has been speculated that CYP2 subfamily diversification occurred principally in response to animal-plant "chemical warfare" (Gonzalez and Nebert, 1990; Nelson, 1998; Danielson, 2002; Nelson, 2003), i.e., to detoxify plant secondary metabolites. All ancestors demonstrated monooxygenation activity towards typical xenobiotic CYP2 substrates using the extant human NADPH-cytochrome P450 reductase (CPR) as a redox partner. In general, the extant forms showed greater turnover of the substrates assessed; however, the activity shown by the ancestors was in the same range as their extant descendants overall. Overall, both the older and younger ancestors showed a significant degree of promiscuity.

Comprehensively quantifying catalytic promiscuity requires resource-intensive screening of catalytic activities towards very large sets of substrates. Therefore, a high-throughput competition assay using a probe substrate for each ancestor was used to determine the ability of ancestors to interact with a large set of commercial pharmaceutical compounds which are typical substrates, inhibitors, and activators of extant P450s. This allowed a quantification of ligand-binding promiscuity using Tanimoto distances to describe the structural and chemical similarity of

ligands. Depending on the ancestral form, activity was affected by more than 20% by ~10–30 of the 47 compounds screened. No significant evolutionary trends were observed across the two major lineages in a quantitative measure of promiscuity with values similar to extant forms (Nath and Atkins, 2008). This suggests that the ancestors are promiscuous in their ability to bind, and potentially also monooxygenate, chemically diverse ligands, similar to their extant descendants. Inhibition may be a poor proxy for catalytic promiscuity since even relatively substrate-specific P450s can be inhibited by diverse chemicals (Nath et al, 2010; Foti et al, 2011). However, the affinity of ancestral forms for several representative ligands was in the typical ~micromolar range seen for interactions of xenobiotic and steroid ligands with extant CYP2 forms. This suggests there has been no loss or gain in overall affinity for such ligands across evolution, consistent with an ancestral detoxification function. Thus, based on the limited number of substrates and ligands assessed here, we speculate that the ancestral CYP2s were also able to catalyze the clearance of compounds of diverse structure and chemical nature, albeit with low catalytic rates.

Extant CYP2 forms metabolize endogenous compounds as well as xenobiotics, mostly to facilitate clearance but sometimes producing bioactive metabolites. Based on the results obtained here, testosterone and other steroids may have been endogenous ancestral substrates, just as they are substrates of extant forms. One or more ancestors may also have been capable of the efficient biotransformation of endobiotics not studied here, e.g., fatty acids, eicosanoids, or other endogenous compounds. In this regard, extant CYP2 forms participate in the metabolism of arachidonic acid (Rifkind et al, 1995), retinoic acid (McSorley and Daly, 2000), and lauric acid (Amet et al, 1994). Indole may also have been an important, "endogenous", ancestral substrate based on its likely absorption from the gut after tryptophan degradation by gut flora. Extant CYP2 forms, especially from the CYP2A and CYP2F subfamilies, have been shown to be more active towards this substrate than CYP1 or CYP3 forms (Gillam et al, 1999; Gillam et al, 2000; Behrendorff et al, 2012). Further studies are ongoing to clarify the potential role of these ancestors in the biotransformation of such compounds and to identify alternative substrates using unbiased metabolomics. However, the likelihood of significant differences between the metabolic biochemistry of primitive tetrapods and any extant animals makes it impossible to speculate about alternative substrates or physiological roles of ancestral CYP2 forms beyond detoxification of lipophilic xenobiotic or endogenous substrates.

The overall picture that emerges here is of relatively promiscuous and highly thermostable basal ancestors that showed low turnover towards a broad range of xenobiotic substrates. These subsequently gave rise to multiple evolutionary lineages that diversified to show increased relative activity towards different subsets of xenobiotic substrates, but still retained a significant degree of promiscuity. The result of this evolutionary diversification is the battery of CYP2 forms we see today, e.g., the ~ten xenobiotic-metabolizing CYP2 forms in humans, that show broad but overlapping substrate specificity and varying, but, at best, moderate, catalytic rates. This conclusion differs somewhat from the observations made about the CYP3 family, where there was little difference in substrate promiscuity or overall extents of activity between the vertebrate ancestor, CYP3_N1, and the major

extant human form, CYP3A4 (Gumulya et al, 2018). Importantly, our observations on both families support the contention that stability and function are not inherently incompatible and that proteins can be stabilized without sacrificing functional plasticity (Giver et al, 1998; Zhao and Arnold, 1999; Komor, 2012). Indeed, P450s may conform to the paradigm of the polarized innovable protein fold advanced by Tawfik and colleagues (Dellus-Gur et al, 2013; Tóth-Petróczy and Tawfik, 2014), where a stable catalytic and structural core is matched with flexible regions involved in substrate binding.

The fact that a stable protein fold can also embody significant promiscuity is of particular interest for protein engineering since robust enzymes are needed for industrial applications. The versatility of the CYP2 and CYP3 families and their dominant role in the metabolism of ~75% of marketed pharmaceuticals (Rendic and Guengerich, 2015) has earned them recognition as potentially useful biocatalysts for the production of drug precursors. Improving thermal tolerance of eukaryotic P450s using traditional methods of engineering, namely site-directed, random, and recombinatorial mutagenesis, has proven challenging (Kumar et al, 2006; Kumar et al, 2007; Talakad et al, 2010). Such attempts have resulted in either loss of thermostability or increases of less than 5 °C, with the greatest improvement to date being an increase in the $T_m$ of CYP2B6 by ~7 °C (Talakad et al, 2010). More success has been achieved in improving the thermostability of prokaryotic P450s, particularly CYP102 forms (Salazar et al, 2003; Otey et al, 2004; Bloom et al, 2006; Otey et al, 2006; Li et al, 2007; Romero et al, 2013). The most stable CYP102 mutant displayed a $^{10}T_{50}$ of 69.7 °C, but was only obtained after several rounds of directed evolution using diverse approaches (Romero et al, 2013). Six of the ancestors resurrected here show $^{10}T_{50}$ values comparable to or greater than this CYP102 mutant. Importantly, the results of the hydrophobicity analysis suggest that P450 thermostability could be enhanced by increasing surface hydrophilicity, an approach that should avoid disturbing active site topology and therefore substrate specificity.

In summary, the results of this study are consistent with the hypothesis that robust protein folds facilitate evolutionary diversification and that the ancestors of the tetrapod CYP2ABGSFTCEH clade were likely to have been able to metabolize a similar variety of xenobiotics as do their modern descendants. Our study suggests that there is no inherent trade-off between stability and promiscuity in a protein fold, and segregation of hydrophobic and hydrophilic residues in the core and surface of a protein may be important in determining stability, providing a strategy for engineering highly stable yet versatile proteins for biotechnology. The thermostability of the CYP2 tetrapod clade appears to have been lost in parallel with the general cooling of global temperatures since ~500 Ma (Ontiveros et al, 2023). Observed stabilities are towards the upper extreme of, or exceed, those expected for vertebrates throughout this period, even accounting for fluctuations in global temperature and latitudinal differences between ecosystems. However, this excess stability and the resulting increased tolerance of mutations may have been important for the rapid evolution of this family of detoxification enzymes in response to animal:plant chemical warfare. ASR of other vertebrate protein families with different physiological roles could reveal whether the trends in thermostability evident for P450s and in other isolated examples (Bar-Rogovsky et al, 2013) are seen more generally in the evolution of other proteins across the same geological timeframe or

whether they are a feature of enzymes that require a high degree of polymorphism to respond rapidly to constantly changing selection pressures such as dietary toxins. Such studies could also provide clues as to the environmental conditions experienced by early vertebrates (Garcia et al, 2017; Ontiveros et al, 2023).

# Methods

### Reagents and tools table

| Reagent/resource | Reference or source | Identifier or catalog number |
|---|---|---|
| **Experimental models** | | |
| *Escherichia coli* strain DH5aF'IQ™ | ThermoFisher Scientific | Cat# 18288019 |
| **Recombinant DNA** | | |
| pCW( + ori) | Gegner and Dahlquist (1991). Backbone of vector available from Addgene in an expression plasmid for 2C9 in pCW( + ori). | Plasmid #69554 |
| pCW/hCPR | Parikh et al, 1997 | |
| pCW/2B6 | Hanna et al, 2000 | |
| pCW/2E1 | Gillam et al, 1994 | |
| pCW/1A2/hCPR | Parikh et al, 1997 | |
| pCW/2A6 | Soucek, 1999 | |
| pCW/2A13 | Shukla et al, 2009 | |
| pCW/2B1 and pCW/2B4 | Saribas et al, 2001 | |
| pCW/2F3 | Wang et al, 1998 | |
| pCW/2C8 pCW/2C9 pCW/2C18 pCW/2C19 | Richardson et al, 1995 | |
| pCW/2C11 | Licad-Coles et al, 1997 | |
| pGro7 | TaKaRa Bio | Cat# 3340 |
| **Antibodies** | | |
| **Oligonucleotides and other sequence-based reagents** | | |
| Genestrings for ancestral P450s | This study, synthesized by GeneArt/Life Technologies/ThermoFisher. | https://doi.org/10.6084/m9.figshare.30908042 |
| **Chemicals, enzymes, and other reagents** | | |
| Gibson Assembly master mix | New England Biolabs | Cat# E2611S |
| Restriction enzymes: NdeI SalI XbaI HindIII | New England Biolabs | Cat# R0111S Cat# R0138S Cat# R0145S Cat# R0104S |
| Isopropyl-β-D-thiogalactoside | Astral Scientific | Cat# AST0487 |
| Yeast extract | ThermoFisher Scientific | Cat# 211929 |
| Bactotryptone | ThermoFisher Scientific | Cat# 211699 |
| Bactopeptone | Amyl Media (Dandenong, Australia) | Cat# RM263 |
| Thiamine | Merck-Sigma-Aldrich (Castle Hill, Australia) | Cat# T4625 |
| L-(+)-Arabinose | Merck-Sigma-Aldrich (Castle Hill, Australia) | Cat# A3256 |
| δ-Aminolevulinic acid | Merck-Sigma-Aldrich (Castle Hill, Australia) | Cat# 08339 |

| Reagent/resource | Reference or source | Identifier or catalog number |
|---|---|---|
| Luminogenic P450-Glo™ kits with probe substrates: luciferin 6'-chloroethyl ether (Luciferin-CEE), luciferin 6'-methyl ether (Luciferin-ME), 6'-deoxyluciferin (Luciferin-H), the ethylene glycol ester of luciferin 6'-methyl ether (Luciferin-ME EGE), the ethylene glycol ester of 6'-deoxyluciferin (Luciferin-H EGE), and luciferin 6'-pentafluorobenzyl ether (Luciferin-PFBE) | Promega Australia | Cat# V8751 Cat# V8771 Cat# V8791 Cat# V8891 Cat# V8881 Cat# V8901 |
| luciferin 6'-methyl ether (Luc-MultiCYP) | Gift of Drs. John Kowalski and Allan Rettie of the University of Washington (Seattle, WA, USA). (Formerly available from Promega Corporation) | Cat# P1731 (Promega) |
| 7-Ethoxyresorufin, 7-methoxyresorufin and 7-pentoxyresorufin | AnaSpec (Fremont, CA, USA) | Cat# AS-85715 Cat# AS-85707 Cat# AS85728 |
| 7-Benzyloxyresorufin | Merck-Sigma-Aldrich (Castle Hill, Australia) | Cat# B1532 |
| Coumarin | Merck-Sigma-Aldrich (Castle Hill, Australia) | Cat# C4261 |
| Androstenedione | Merck-Sigma-Aldrich (Castle Hill, Australia) | Cat# SML2364 |
| Testosterone | Merck-Sigma-Aldrich (Castle Hill, Australia) | Cat# T1500 |
| Progesterone | Merck-Sigma-Aldrich (Castle Hill, Australia) | Cat# 5341 |
| 2α-Hydroxytestosterone, 2β-hydroxytestosterone, 6α-hydroxytestosterone, 6β-hydroxytestosterone, 15β-hydroxytestosterone, 16α-hydroxytestosterone, 16β-hydroxytestosterone | Steraloids Inc. (Newport, RI, USA) | Cat# B1356 Cat# B1210 Cat# B1424 Cat# B1830 Cat# C212 Cat# B1014 Cat# L1524 |
| 15α-Hydroxytestosterone | Hunter et al, 2011 | |
| Acetaminophen, aminopyrine, amlodipine, amodiaquine, atorvastatin, benzbromarone, carbamazepine, celecoxib, chlorzoxazone, clopidogrel, clozapine, dextromethorphan, diclofenac, diltiazem, donepezil, ethinylestradiol, erythromycin, fluoxetine, flutamide, furosemide, indomethacin, imipramine, levofloxacin, metoprolol, midazolam, nevirapine, olanzapine, omeprazole, paclitaxel, pioglitazone, pravastatin, procainamide, propranolol, ritonavir, rosiglitazone, sulfamethoxazole, tacrine, tamoxifen, ticlopidine, tienilic acid, valproic acid, valsartan, verapamil, warfarin (racemic), R-warfarin, S-warfarin, zafirlukast and zomepirac | Compound Management, AstraZeneca R&D | |
| **Software** | | |
| GRASP | Foley et al, 2022 | |
| MAFFT | Katoh et al, 2002 | |
| T-REX web server | Boc et al, 2012 | |
| PhyML | Guindon and Gascuel, 2003 | |
| BioNJ | Gascuel, 1997 | |
| GeneOptimizer™ | GeneArt/ThermoFisher Scientific; Raab et al, 2010 | |

| Reagent/resource | Reference or source | Identifier or catalog number |
|---|---|---|
| NUPACK | Zadeh et al, 2011 | |
| mRNA Optimizer | Gaspar et al, 2013 | |
| Open Babel | O'Boyle et al, 2011 | |
| Other | | |

## Materials

The expression vectors for extant P450s were provided by Prof. F.P. Guengerich (pCW/2B6 (Hanna et al, 2000), pCW/2E1 (Gillam et al, 1994) and pCW/1A2/hNPR (Parikh et al, 1997)), Dr. Pavel Soucek (pCW/2A6 (Soucek, 1999)) Dr Imad Hanna (Schering-Plough Research Institute, Kenilworth, NJ, USA; pCW/2A13) and Prof. Garold S. Yost (University of Utah, Salt Lake City, UT, USA; pCW/2F3 (Wang et al, 1998)) or were constructed from cDNAs provided by Prof. Philippe Beaune (INSERM U490, Universite René Descartes, Paris-V, France; CYP2C18 and CYP2C19), Prof. D.J. Birkett (Flinders University of South Australia; CYP2C8 and CYP2C9), Dr. Lucy Waskell (University of Michigan, MI; CYP2B1 and CYP2B4 (Saribas et al, 2001)) and Dr. Almira Correia (pCW/2C11). P450s were co-expressed with human NADPH-dependent P450 reductase (hCPR) in bicistronic format as described previously (Parikh et al, 1997; Gillam et al, 1999; Cuttle et al, 2000; Gillam et al, 2000; Notley et al, 2002; Kinobe et al, 2005). The expression vector encoding the GroEL/GroES chaperone system (pGro7) was donated by Professor K. Nishihara (HSP Research Institute, Kyoto, Japan).

Oligonucleotides and genestrings were synthesized by GeneArt/Life Technologies/ThermoFisher (Australia). Restriction endonucleases, Gibson Assembly master mix, and other DNA-modifying enzymes were purchased from New England Biolabs (Notting Hill, Australia). The *E. coli* strain DH5α F'IQ was obtained from Life Technologies. Yeast extract and Bactotryptone for culture media were purchased from ThermoFisher (Australia) and peptone from Amyl Media (Dandenong, Australia). Thiamine, L-(+)-arabinose, and δ-aminolevulinic acid were purchased from Merck-Sigma-Aldrich and isopropyl-β-D-thiogalactoside (IPTG) from Astral Scientific (Taren Point, Australia). Luminogenic P450-Glo™ kits with probe substrates luciferin 6'-chloroethyl ether (Luc-CEE), luciferin 6'-methyl ether (Luc-ME), 6'-deoxyluciferin (Luc-H), the ethylene glycol ester of luciferin 6'-methyl ether (Luc-ME EGE), the ethylene glycol ester of 6'-deoxyluciferin (Luc-H EGE), and luciferin 6'-pentafluorobenzyl ether (Luc-PFBE) were obtained from Promega (Alexandria, Australia). The methyl ester of luciferin 6'-methyl ether (Luc-MultiCYP) was the generous gift of Drs. John Kowalski and Allan Rettie of the University of Washington (Seattle, WA, USA).

Coumarin, testosterone, progesterone, androstenedione, and 7-benzyloxyresorufin were obtained from Sigma-Aldrich. 7-Ethoxyresorufin, 7-methoxyresorufin, and 7-pentoxyresorufin were purchased from AnaSpec (Fremont, CA, USA). 2α-Hydroxytestosterone, 2β-hydroxytestosterone, 6α-hydroxytestosterone, 6β-hydroxytestosterone, 15β-hydroxytestosterone, 16α-hydroxytestosterone, and 16β-hydroxytestosterone were purchased from Steraloids (Newport, RI, USA). Androstenedione was purchased from Sigma Merck (Castle Hill, Australia). 15α-Hydroxytestosterone was

isolated from extracts of incubations with ancestral CYP2A enzyme and identified by NMR using previously described methods (Hunter et al, 2011). Acetaminophen, aminopyrine, amlodipine, amodiaquine, atorvastatin, benzbromarone, carbamazepine, celecoxib, chlorzoxazone, clopidogrel, clozapine, dextromethorphan, diclofenac, diltiazem, donepezil, ethinylestradiol, erythromycin, fluoxetine, flutamide, furosemide, indomethacin, imipramine, levofloxacin, metoprolol, midazolam, nevirapine, olanzapine, omeprazole, paclitaxel, pioglitazone, pravastatin, procainamide, propranolol, ritonavir, rosiglitazone, sulfamethoxazole, tacrine, tamoxifen, ticlopidine, tienilic acid, valproic acid, valsartan, verapamil, warfarin (racemic), R-warfarin, S-warfarin, zafirlukast and zomepirac were supplied by Compound Management, AstraZeneca R&D. All other reagents were obtained from local suppliers at the highest quality available.

## ASR of the tetrapod clade of the CYP2 family

A total of 975 CYP2 sequences were collected from UniProt, NCBI, and the cytochrome P450 homepage (Nelson, 2009). The sequences were aligned using MAFFT (Katoh et al, 2002). The alignment was curated manually with reference to a previous phylogenetic study (Kirischian et al, 2011) to improve reliability at the gap positions. Phylogenetic relationships of these CYP2 sequences were reconstructed via the T-REX web server (Boc et al, 2012), using the maximum likelihood (ML) method under the Jones–Taylor–Thornton (JTT) substitution model (Jones et al, 1992) with PhyML (Guindon and Gascuel, 2003). The default parameters were used, and the initial tree was determined by neighbor-joining (BIONJ) (Gascuel, 1997). The final tree was inferred using PhyML. Bootstrapping analysis with 100 replicates was performed to evaluate the tree. The tree was rooted using the CYP2M subfamily (found in Actinopterygii) as an outgroup. Sequences of all internal (ancestral) nodes of the constructed phylogenetic tree were inferred via GRASP (Foley et al, 2022) using joint reconstruction. For comparative purposes, marginal reconstructions were inferred via GRASP at each node of the same tree allowing calculation of the posterior probabilities of all residues at each position of the alignment and derivation of an 'AltAll' ancestor, in which the second most probable residue was substituted at any position of the marginal sequence where the second most probable residue had a posterior probability of 0.4 or greater.

## Optimization and cloning of ancestral DNA sequences

The 5' end of the predicted ancestral sequences, encoding the N-terminal membrane anchor, was modified to enhance bacterial expression. Codons upstream of the conserved poly-proline motif, PPGP, present in all reconstructed ancestral sequences, were replaced with the sequence encoding an MAKKTSSKGKL leader sequence (von Wachenfeldt et al, 1997). A hexa-His tag was fused to the C-terminus of the ancestral P450 coding sequence by a Ser–Thr linker encoding a SalI site. The inferred ancestral DNA sequences downstream of the N-terminal modification were codon-optimized to match *E. coli* codon preferences using GeneOptimizer™ (Raab et al, 2010) available through GeneArt™ (ThermoFisher). The sequence of the inferred mRNA encompassing residues −20 to +96 (with respect to the start codon) was assessed for potential

secondary structure formation using NUPACK (Zadeh et al, 2011), and the predicted $\Delta G$ for secondary structure formation was increased using mRNA Optimizer (Gaspar et al, 2013) and manually introducing silent mutations. The modified ancestral cDNA sequences were synthesized with ~60 additional nucleotides at the 5' and 3' ends for Gibson Assembly (Gibson et al, 2009) into the vector backbone from pCW/1A2/hCPR (Parikh et al, 1997) cut with NdeI and XbaI to excise the CYP1A2 open reading frame (ORF). An exception to this was the CYP2ABGSFTCEH ancestor, which was obtained from the supplier already subcloned in the pMA-RQ plasmid, then excised using NdeI and XbaI and ligated into the similarly digested backbone from the pCW/1A2/hNPR vector. Constructs were sequenced throughout the ORFs by automated dideoxy sequencing at the Brisbane Node of the Australian Genome Research Facility (Brisbane, Australia).

## Expression and thermostability analysis of recombinant P450s

P450s were expressed in *E. coli* according to published procedures (Gillam et al, 1993; Notley et al, 2002) and P450 holoprotein was quantified by Fe(II).CO *vs*. Fe(II) difference spectroscopy (Johnston and Gillam, 2013). Thermostability was assessed as described previously (Gumulya et al, 2018). The residual folded protein was converted to a proportion of the total amount of P450 measured in an unheated sample maintained at 25 °C. The temperature at which 50% of the P450 retained the Fe(II).CO *vs*. Fe(II) difference spectrum characteristic of an intact enzyme after a 60 or 10 min incubation ($^{60}T_{50}$ and $^{10}T_{50}$, respectively) was interpolated from residual P450 *vs*. temperature plots fitted with a sigmoidal curve, in GraphPad Prism.

## Homology modeling

Homology models were constructed using the program Modeller (Fiser and Sali, 2003) via the graphical interface Chimera (Pettersen et al, 2004). For each enzyme, a BLAST search was carried out on the PDB database to find a suitable structural template based on shared sequence identity. The query and template sequences were aligned, and the homology modeling tool in Modeller was applied. For each round of model building, Modeller returned five possible model iterations. The model chosen for analysis was that with the lowest normalized Discrete Optimized Protein Energy score.

## Analysis of hydrophobicity and residue interaction networks

Residues of all analyzed forms were categorized as either "buried" or "exposed" based on the solvent exposure of the equivalent residue in crystal structures of forms CYP2A6 (PDB: 1Z10), CYP2B6 (3IBD) and CYP2C9 (1OG2), using the following criteria: Swiss-PDB viewer was used to identify residues that were >30% solvent accessible in at least one of the crystal structures, and these residues were classified as "exposed"; residues that were ≤30% solvent accessible were classified as "buried". The hydrophobicity of each residue was calculated according to the Kyte–Doolittle hydropathy index (Kyte and Doolittle, 1982). A phylogenetic independent contrast analysis (Garland et al, 1992) was performed on the twelve extant sequences to identify any correlation between

the buried/exposed values and thermostability, while accounting for the phylogenetic structure.

Models of each ancestral P450 were also constructed using AlphaFold 2 (Jumper et al, 2021; Evans et al, 2022) via the Galaxy Server (The Galaxy Community, 2024). Interactions between residues in each ancestor were computed in the best model using the RING 4.0 server (Del Conte et al, 2024), and networks were compared among the ancestors, focusing on the three oldest ancestors CYP2ABGSFTCEH, CYP2CEH, and CYP2ABGSFT (95–98% identical; Appendix Table S1).

## Analysis of activity towards luminogenic and fluorogenic substrates

Activities were measured using bacterial membranes prepared from cells co-expressing the P450 of interest and human CPR according to established procedures in microplates (Chang and Waxman, 2006). P450 and CPR content was quantified as described (Guengerich, 1994). Reactions with luciferin derivatives were carried out as described previously (Huang et al, 2007; Johnston et al, 2007) except that P450 concentrations were normalized to 40 µM, and substrates were used in the following concentrations: 5 µM luciferin-H EGE; 15 µM luciferin-CEE and luciferin-ME EGE; 25 µM luciferin-PFBE; 50 µM luciferin-ME and luciferin-H; and 100 µM Luciferin-MultiCYP. Alkoxyresorufin O-dealkylation and coumarin hydroxylation were measured as described previously (Kim et al, 2005; Chang and Waxman, 2006; Waxman and Chang, 2006) using 0.1 µM P450 and 5 µM of alkoxyresorufin or 50 µM coumarin. Incubations were initiated by the addition of an NADPH-generating system (NGS) consisting of 10 mM glucose-6-phosphate, 250 µM NADP$^+$ and 0.5 U.ml$^{-1}$ glucose-6-phosphate dehydrogenase and quenched at the times indicated for individual experiments.

## Testosterone hydroxylation assays

Screening for testosterone metabolism was performed using HPLC as described previously (Hunter et al, 2011). Reactions were carried out at a testosterone concentration of 100 µM using 0.5 µM P450 delivered in membranes from *E. coli* co-expressing P450 and hCPR. Metabolites were identified by comparison of retention times with authentic standards of 2α-hydroxytestosterone, 2β-hydroxytestosterone, 6α-hydroxytestosterone, 6β-hydroxytestosterone, 15α-hydroxytestosterone, 15β-hydroxytestosterone, 16α-hydroxytestosterone, 16β-hydroxytestosterone, and androstenedione and quantified with reference to standard curves prepared with the corresponding metabolite standard.

## Competitive binding assay

Each ancestor was screened for binding to 48 drug compounds by assessing their effect on activity towards a marker substrate in a competition assay. Marker substrates used for screening were: coumarin for CYP2A; luciferin-multiCYP for CYP2C, CYP2E, CYP2F, CYP2CE, CYP2ABGS, CYP2CEH, CYP2ABGSFT and CYP2ABGSFTCEH; 7-benzyloxyresorufin for CYP2B; 7-ethoxyresorufin for CYP2S; and 7-methoxyresorufin for CYP2BS. The substrate concentration used in each screen was equal to or less than the $K_{0.5}$ established for each form. Coumarin reactions were

carried out according to established protocols (Kim et al, 2005; Waxman and Chang, 2006) using 50 nM P450 and 1 µM substrate. Luciferin-MultiCYP reactions were carried out in a total volume of 25 µL in Nunc® 384-well plates, with an enzyme concentration of 40 nM and respective $K_{0.5}$, for 40 min at 37 °C. Alkoxyresorufin reactions were carried out as described previously (Chang and Waxman, 2006) using 100 nM of CYP2B and CYP2S, and 200 nM of CYP2BS. Due to the limited solubility of 7-benzyloxyresorufin, the true $K_{0.5}$ for CYP2B could not be established; therefore, a concentration of 5 µM, significantly lower than the expected $K_{0.5}$, was used for screening with this substrate. 7-Ethoxy- and 7-methoxyresorufin were used at concentrations of 1.5 and 2.0 µM, respectively. The effectors were added to a final concentration of 50 µM.

## Calculation of Tanimoto distances and analysis of ligand promiscuity

The ligand set of any given enzyme was defined by compounds that activated or inhibited the P450s activity by >20%. Each of the 48 compounds used in the ligand screen was assigned a binary molecular fingerprint based on the MACCS keyset of 166 chemical descriptors using Open Babel (O'Boyle et al, 2011). For each pair of compounds, the molecular similarity was defined by the Tanimoto coefficient ($T_{ab}$) (Tanimoto, 1958), i.e., for a pair of compounds A and B, where $a$ is the number of features present only in A, $b$ is the number of features present only in B, and $c$ is the number of features present in both A and B, the Tanimoto coefficient is defined as $T_{ab} = \frac{c}{a+b-c}$. The Tanimoto coefficients of all the compounds in the ligand set of a given enzyme were averaged. Ligand binding promiscuity was quantified as described previously (Nath and Atkins, 2008; Foti et al, 2011) except that the relative change in activity was used in place of $k_{cat}/K_M$ values as a semiquantitative measure of the interaction of ligands with ancestral enzymes to obtain effector promiscuity ($J_{eff}$) values rather than inhibitory promiscuity ($J_{inh}$) values. Where ligands increased rather than inhibited the enzyme activity, the relative change in activity was calculated as the activity of the ligand-free control divided by that seen in the presence of effector, converted to a percentage so that the relative impact of the effect used for the calculation was agnostic to its direction (i.e., inhibition *vs.* stimulation of activity).

## Inference of consensus proteins

For each subgroup of sequences corresponding to the descendants of a given resurrected ancestor, the corresponding consensus protein was generated using the EMBOSS Cons tool on the European Bioinformatics Institute web server. Pairwise alignments were run using the EMBOSS Needle tool to determine sequence identity and similarity to the inferred ancestors.

## Data availability

The data that support the findings of this study are available in the supporting information or upon request from the corresponding authors via the FigShare repository: https://doi.org/10.6084/m9.figshare.30436768.

The source data of this paper are collected in the following database record: biostudies:S-SCDT-10_1038-S44318-026-00699-y.

## Peer review information

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

## Acknowledgements

The authors thank Dr. Abhinav Nath (University of Washington, Seattle) for assistance and advice concerning the promiscuity analysis; Prof David R Nelson for advice on some phylogenetic relationships; Prof Allan Rettie and Dr. John Kowalski (University of Washington, Seattle) for providing luciferin MultiCYP; and Drs. FP Guengerich, Pavel Souček, Imad Hanna, Garold S Yost, Philippe H Beaune, Donald J Birkett, Lucy Waskell, M Almira Correia and K Nishihara for donating plasmids. Funding for this work was from Australian Research Council Discovery Project Grants DP120101772 and DP160100865 and AstraZeneca Innovative Medicines and Early Development, Cardiovascular, Renal and Metabolism, Gothenburg. REST, CMR, and GF were supported by Australian Postgraduate Research Training Awards.

## Author contributions

**Raine E S Thomson**: Conceptualization; Data curation; Formal analysis; Investigation; Visualization; Methodology; Writing—original draft; Writing—review and editing. **Yosephine Gumulya**: Conceptualization; Data curation; Formal analysis; Supervision; Investigation; Methodology; Writing—review and editing. **Anthony W Bengochea**: Formal analysis; Methodology; Writing—review and editing. **Gabriel Foley**: Data curation; Software; Methodology; Writing—review and editing. **Julian Zaugg**: Software; Formal analysis; Methodology; Writing—review and editing. **Anna Aagaard**: Formal analysis; Supervision; Investigation; Methodology; Writing—review and editing. **Connie M Ross**: Resources; Formal analysis; Writing—review and editing. **James Beckett**: Investigation; Methodology. **Mikael Bodén**: Supervision; Methodology; Writing—review and editing. **Ulrik Jurva**: Data curation; Formal analysis; Supervision; Investigation; Writing—review and editing. **Martin A Hayes**: Resources; Supervision; Project administration; Writing—review and editing. **Shalini Andersson**: Resources; Supervision; Funding acquisition; Project administration; Writing—review and editing. **Elizabeth M J Gillam**: Conceptualization; Data curation; Formal analysis; Supervision; Funding acquisition; Visualization; Methodology; Writing—original draft; Project administration; Writing—review and editing.

Source data underlying figure panels in this paper may have individual authorship assigned. Where available, figure panel/source data authorship is listed in the following database record: biostudies:S-SCDT-10_1038-S44318-026-00699-y.

## Disclosure and competing interests statement

Some of the thermostable xenobiotic-metabolizing P450s developed in this work have been included in an Australian Provisional Patent Application (#2015902345, PCT/AU2016/050449) and licensed for application as biocatalysts in pharmaceutical and fine chemical synthesis under the tradename "CYPerior". Authors Aagaard, Jurva, Hayes, and Andersson were employees of AstraZeneca (Sweden) during the time this study was conducted.

# Expanded View Figures

**Figure EV1.  CYP2 phylogenetic tree.**

The CYP2 clade used for reconstruction incorporates the tetrapod subfamilies, CYP2A, CYP2B, CYP2C, CYP2E, CYP2F, CYP2G, CYP2H, CYP2S and CYP2T, and was inferred from 975 extant CYP2 sequences. Sequences were collected from NCBI and Uniprot databases using a BLAST-search of sequences with >40% similarity to characterized CYP2 forms. Phylogenetic relationships were reconstructed via the T-REX web server (Boc et al, 2012), using the maximum likelihood (ML) method under the Jones–Taylor–Thornton (JTT) substitution model (Jones et al, 1992) with PhyML (Guindon and Gascuel, 2003). The final tree was inferred using PhyML. Bootstrapping analysis with 100 replicates was performed to evaluate the tree. The CYP2M subfamily was used as an outgroup to root the tree. Nodes are labeled with bootstrap values. Extant forms are labeled with a five letter code comprising the first three letters of the genus and two letters of the species name, e.g., HOMSA for *Homo sapiens*.

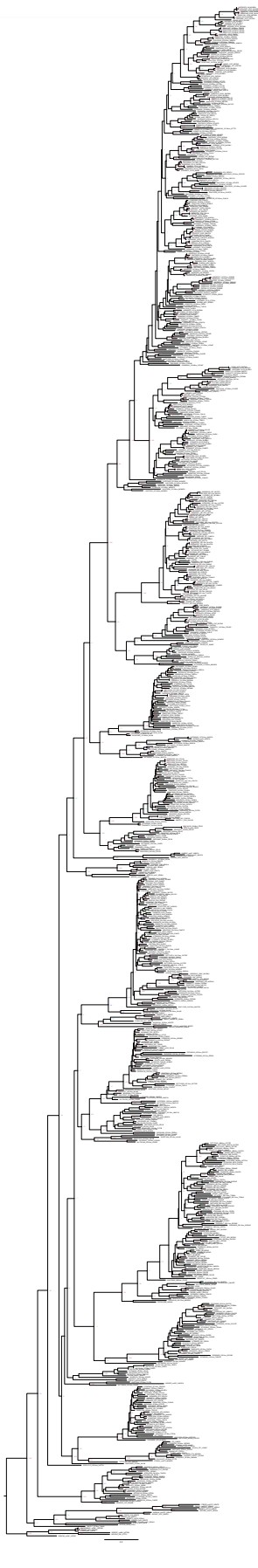

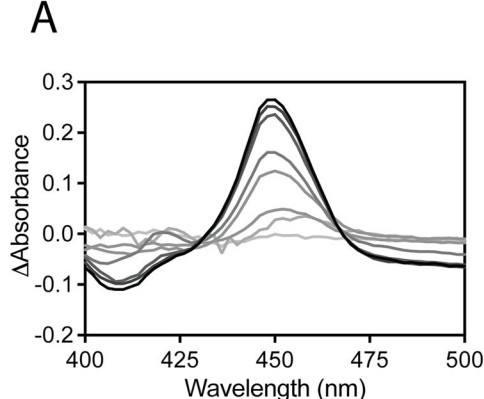

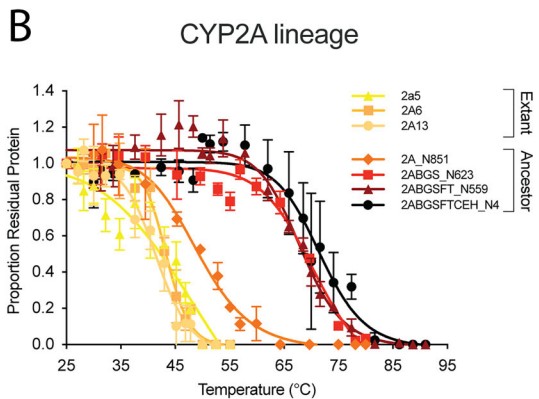

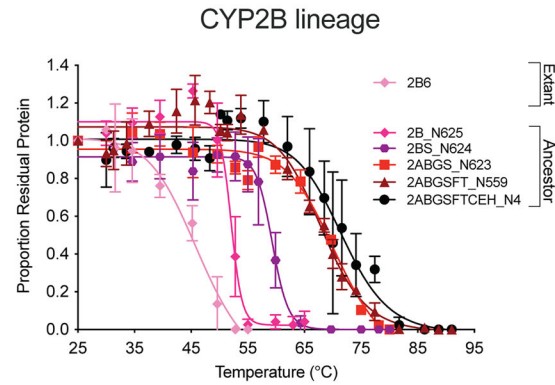

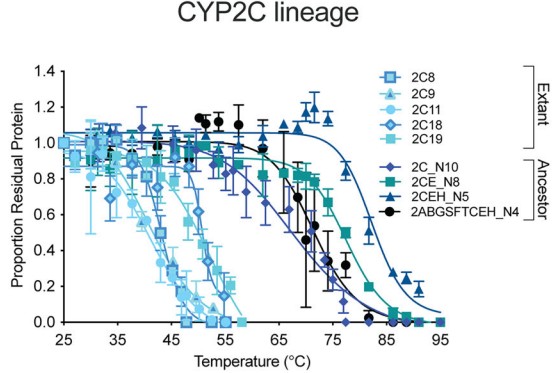

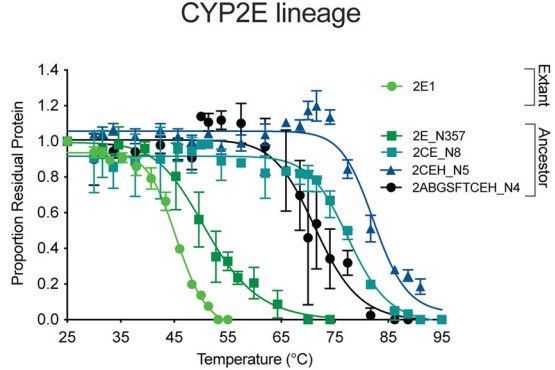

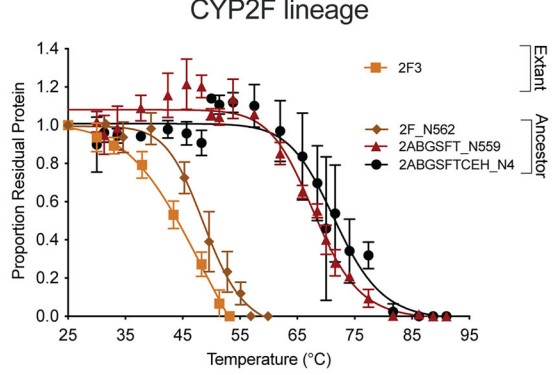

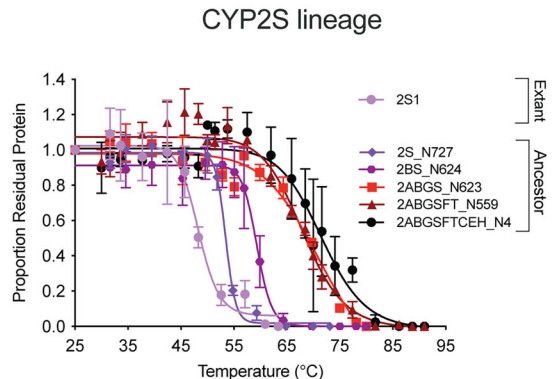

◄   **Figure EV2.  Thermal stability profiles of extant and ancestral variants of CYP2 enzymes.**

Bacterial cells expressing each enzyme were heated for 60 min at temperatures between 25 °C and 100 °C, followed by 5 min at 4 °C and equilibration to room temperature for 5 min. The P450 content was then measured by Fe(II).CO *vs.* Fe(II) difference spectroscopy. (**A**) Representative spectra show the loss of the peak at 450 nm in samples heated at increasing temperatures from 25 °C (black) to 65 °C (light gray). (**B**) The P450 remaining intact in heated samples was calculated as a proportion of total P450 in a sample left at 25 °C. Data represent the mean ± SD of $n = 3$ biological replicates. Each set of plots compares the forms that are related along a single lineage as indicated. All ancestors showed significantly different $^{60}T_{50}$ values to their immediate ancestor along the same lineage except CYP2ABGSFT (compared to CYP2ABGSFTCEH) and CYP2ABGS (compared to CYP2ABGSFT); two-tailed, heteroscedastic Student's *t* test. All extant enzymes showed $^{60}T_{50}$ values that were significantly lower than all their cognate ancestors ($P < 0.006$, two-tailed heteroscedastic Student's *t* test). A detailed statistical analysis of the significance of differences in $^{60}T_{50}$ values can be found in Appendix Table S3.

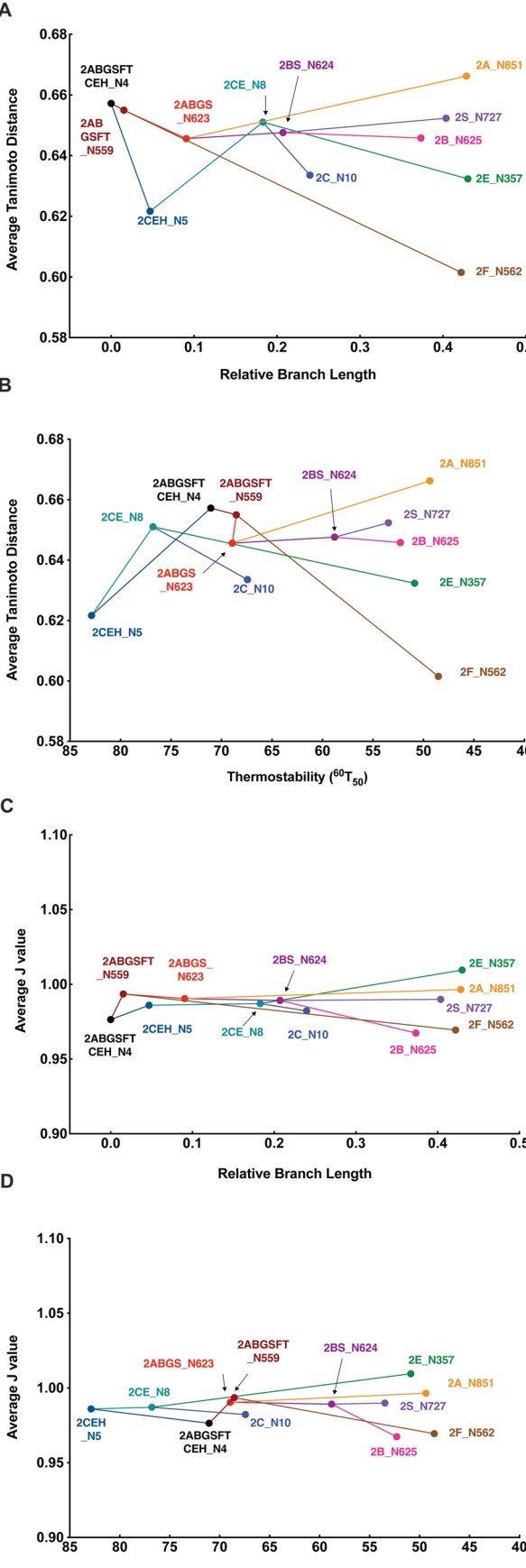

**Figure EV3.  Ligand promiscuity of CYP2 ancestors measured by average Tanimoto distances and *J* values.**

The ligand set of any given enzyme was defined by compounds that activated or inhibited the P450's activity by >20% (Appendix Fig. S8 and Appendix Table S5). The similarity between all compounds in the ligand set was quantified using the Tanimoto distance measure, i.e., for a pair of compounds A and B, where $a$ is the number of features present only in A, $b$ is the number of features present only in B, and $c$ is the number of features present in both A and B, the Tanimoto distance is defined as $T_{ab} = \frac{c}{a+b-c}$. The chemical features of each compound were defined by the MACCS keyset of 166 chemical descriptors. The distance measure between any two compounds was defined as $1 - T_{ab}$ such that greater dissimilarity between two compounds equated to a value closer to 1. The ligand promiscuity of each enzyme was quantified as described in Nath and Atkins (2008) and Foti et al (2011) except that the effect of an inhibitor or activator on a given form was quantified as the proportional change in activity in the presence of the effector, independent of the direction of the effect. (**A**) shows the relationship between average Tanimoto distance and the evolutionary distance between forms (relative branch length). (**B**) shows the relationship between average Tanimoto distance and thermostability ($^{60}T_{50}$). (**C**) shows the relationship between average J value and the evolutionary distance between forms (relative branch length). (**D**) shows the relationship between average J value and thermostability ($^{60}T_{50}$).

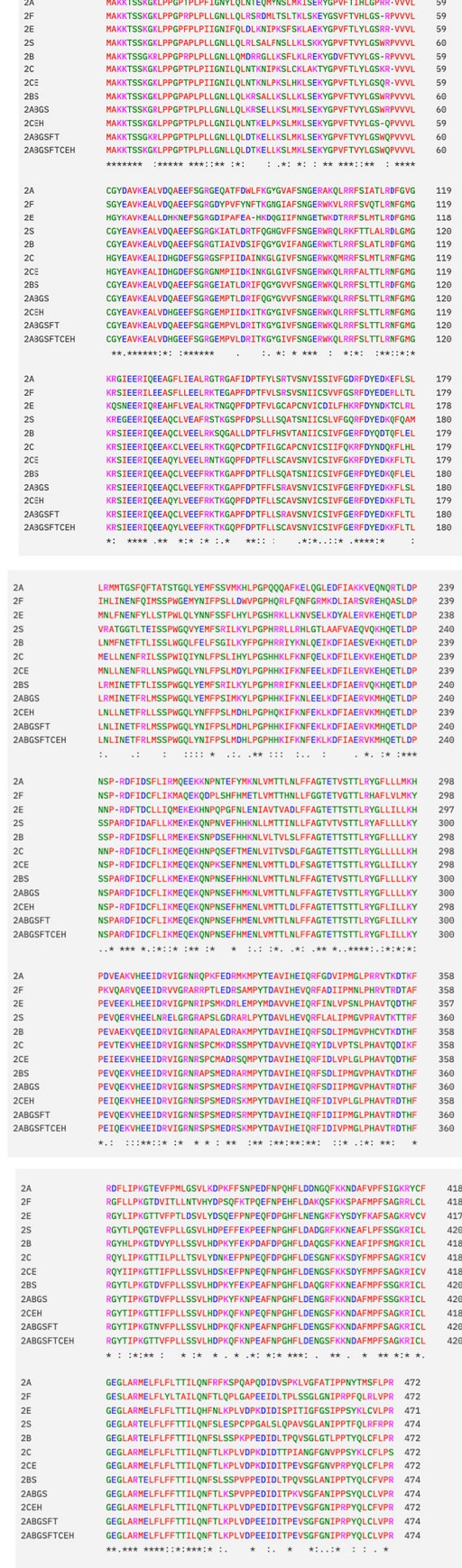

◄ **Figure EV4. Alignment of the ancestral CYP2 sequences resurrected herein.**

Sequences are shown in the form in which they were characterized, including the modifications made to the N-terminal membrane anchor to enable expression in *E. coli*.

