## [Peer Review File · The EMBO Journal]

Thermostable ancestors enabled evolutionary diversification of promiscuous chemical defence enzymes

Raine Thomson, Yosephine Gumulya, Anthony Bengochea, Gabriel Foley, Julian Zaugg, Anna Aagaard, Connie Ross, James Beckett, Mikael Bodén, Ulrik Jurva, Martin Hayes, Shalini Andersson, and Elizabeth Gillam

Corresponding author: Elizabeth Gillam (e.gillam@uq.edu.au)

Review Timeline:

Submission Date:	3rd Mar 25
Editorial Decision:	17th May 25
Revision Received:	10th Oct 25
Editorial Decision:	28th Oct 25
Revision Received:	18th Dec 25
Accepted:	9th Jan 26

Editor: Yehu Moran

Transaction Report:

Dear Prof. Gillam,

Thank you for submitting your manuscript for consideration by the EMBO Journal. It has now been seen by three referees whose comments are shown below.

Given the referees' positive recommendations, I would like to invite you to submit a revised version of the manuscript, addressing the comments of all three reviewers. I should add that it is EMBO Journal policy to allow only a single round of revision, and acceptance of your manuscript will therefore depend on the completeness of your responses in this revised version.

We strongly recommend that after consulting with your co-authors you will send us via email in the next few weeks a revision plan. This should cover any additional experiments and/or analyses you plan to include in your revision. Moreover, if some suggestions by the referees are impossible to perform due to technical limitations or expected to take too long (our standard revision time frame is 3 months) this would be a good opportunity to mention these challenges. From our experience sending us the revision plan in advance will help both you and us to set our expectations and can facilitate a smoother revision process.

Thank you for the opportunity to consider your work for publication. I look forward to your revision.

Yours sincerely,

Yehu Moran
Academic Editor
The EMBO Journal

We realize that it is difficult to revise to a specific deadline. In the interest of protecting the conceptual advance provided by the work, we recommend a revision within 3 months (15th Aug 2025). Please discuss the revision progress ahead of this time with the editor if you require more time to complete the revisions.

Referee #1:

This paper describes the resurrection of ancestral cytochrome P450 enzymes and finds that these enzymes were relatively promiscuous and thermostable. The authors conclude that high thermostability is a driver for subsequent diversification.

While I find this idea generally interesting, the paper makes some questionable methodological choices (see below). I am also not sure that the data presented here is sufficient to establish thermostability as a cause of diversification. In effect, the authors argue that substitution rates later in the tree are to a very large degree caused by the thermostability of ancestors. The statistical evidence for this claim is weak (indeed, no formal analysis is carried out at all to substantiate this), so in my view, this conclusion is speculation, which the authors are, of course, entitled to.

As this paper is concerned with the thermostability of ancestral proteins, it has to deal with the debate around whether these very high stabilities (in some cases 70 degrees Celsius and above) are realistic. Such high melting temperatures are now reserved for proteins from thermophilic organisms that live in niches much hotter than what we can reasonably assume ancestral vertebrates to have experienced in the ocean. One particularly striking example is ancestor 2C, which, as far as I can tell, existed in an ancestral mammal - presumably an endothermic organism that regulated its body temperature. This protein is reconstructed to have a melting point of over 60 degrees Celsius. This puts it far outside the distribution of melting points in vertebrates (according to this recent paper <https://www.nature.com/articles/s41592-020-0801-4>).

The authors argue that such stabilities are a remnant of a time when surface temperatures were even higher, and that stability only falls very slowly due to drift. I find this argument unconvincing. First, it is unclear to me from what time prior to the Cambrian the authors even believe these proteins inherited their extreme stabilities - could this perhaps be stated explicitly? But more importantly, this theory seems at odds with overwhelming biochemical evidence that most mutations are destabilizing. In the absence of continuous selection for stability, proteins should therefore evolve towards the minimal stability they can function with relatively quickly (I am sure the authors are aware of Richard Goldstein's arguments). At the very least, the authors have to wrestle with this inconsistency in their theory. As it stands, the argument feels like motivated reasoning to explain away otherwise extremely puzzling reconstructed stabilities.

I want to be clear here that I know that the senior author's group stands on one side of this argument about the stability of ancestral proteins and this reviewer stands on the other. I have no intention to grandstand in this review process, but I think the paper deserves a frank discussion of the plausibility of these extremely high stabilities. As I explain below, the paper would have benefited from quantifying reconstruction uncertainty.

Apart from these conceptual issues, I have a few methodological questions.

I am puzzled by the unjustified use of joint reconstructions. This is not standard in the field. The reason is that in a joint reconstruction, errors in the reconstructions deep in the tree propagate through all nodes in the tree. In a marginal reconstruction, each node is reconstructed whilst integrating over the uncertainty at all other nodes, meaning that such propagation of reconstruction errors is much less of a problem. Why did the authors choose a joint reconstruction?

In addition, the model is incompletely specified. The substitution matrix is JTT, but what about the state frequencies? Is rate heterogeneity accounted for using a gamma distribution? How was this model chosen? A quick glance at the T-REX web interface seems to suggest that no model finding routine is implemented there.

The same goes for the reconstruction with GRASP. This program's chief advantage is a somewhat better way to assign gaps. It comes with the very considerable disadvantage of only being able to incorporate rate heterogeneity when the alpha parameter

of a gamma distribution is externally supplied. The authors do not state whether this was done here, so I suspect that it wasn't. At the very least, the use of GRASP in this case would have to be carefully justified, as a much more appropriate model of evolution (i.e., one that models rate heterogeneity among sites) was traded in for an uncertain advantage in automated gap assignments.

I could find no information about the quality of the reconstructions in terms of their posterior probabilities. This information has to be provided, and the tree, alignments, and reconstructions need to be deposited somewhere public. Similarly, it appears that the authors have not tested for the effect of statistical (or indeed topological) uncertainty on the exact sequences and properties of their ancestors. This is very disappointing and not up to the standard of the field. Especially for quantitative parameters like stability, this should absolutely be done.

I could not clearly assess the plausibility of the trees, as the trees are not labeled in a way that allows the reader to compare them to known species relationships and whole genome duplications in the vertebrates. There are no branch supports on the trees, and a phylogeny with all taxa labeled is not available in the supplemental information. I also would strongly discourage the authors from using the circular representation of their tree. This representation makes the branching relationships very hard to see and makes it impossible to compare the relative ages of the nodes in terms of branch length. All this representation does is to emphasize that the authors sampled more sequences from the 2C clade than from any other clade.

I don't understand why the authors chose to use relative branch lengths as a measure of age. The divergence times of most vertebrate groups have been estimated many times over and are freely available on TimeTree. On a vertebrate protein phylogeny, it should be simple enough to assign each node to either a speciation node on the species tree, or an interval between speciation nodes for nodes that correspond to gene duplications on the gene tree. Using branch lengths as a proxy for age tacitly assumes that these genes evolve in a clock-like manner, which I highly doubt. In addition, the whole justification for the plausibility of these extremely high melting temperatures is that these proteins are thought to exist at times when the ocean temperature was high. It therefore seems essential to me that the reader can assess that these nodes are really that old. It would also help to plot ocean temperatures as a function of time somewhere in this manuscript.

Figure 3 purports to report a correlation between hydrophobicity and thermostability. The analysis provides only a visual representation of this supposed correlation and no statistical analysis. If the authors want to prove this correlation, they need to carry out an appropriate phylogenetic statistical test (like phylogenetic independent contrasts)

Referee #2:

Review of EMBOJ-2025-120404

Thermostable ancestors enabled evolutionary diversification of promiscuous chemical defense enzymes (Raine et al)
This superb and interesting paper characterizes the ancestrally reconstructed cytochrome P450 family 2 enzymes. These are xenobiotic (and endogenous compound) metabolizing enzymes with a broad range of substrate diversity. Multiple different CYP2 enzymes have evolved in vertebrates, and this research has focused on the more 'recent' tetrapod evolutionary set (that is, those enzymes that have diverged in the last 380 million years).

This work is really fascinating - it touches on protein function and structure relationships, with important implications for industrial applications, as well as very interesting questions about diversity and stability in protein evolution, and the evolution of organisms across hundreds of millions of years.

The paper is very clearly and compellingly written, and I have not found any typographical errors. The methods are well described and justified. The ASR is performed with one of the state of the art methods (GRASP), that takes into account indels, although there are few in the collection. This is not a structural modelling paper, so the use of only 5 models for each homology model is fine. The conclusions are careful, and carefully justified. A few suggestions, and discussion points of note:

- (a) the oldest ancestors were the least well expressed - are there any obvious amino acid composition differences? Do these ancestors have relative expression levels comparable to the CYP1 or CYP3 families, that have similarly been reconstructed?
- (b) Could the plot in Figure 2 have approximate divergence ages as an additional x axis, e.g. based on Timetree?
- (c) One method of examining the range of ASR possibilities is to express the 'alt_all' sequence, that incorporates all the lower probability amino acids at ambiguously reconstructed positions (Eike et al *Mol Biol Evol.* 2016 Oct 30;34(2):247-261. doi: 10.1093/molbev/msw223). How does the alt_all ML reconstruction compare to the consensus reconstruction?
- (d) This family reconstruction encompasses both homeotherms (evolving separately in amniotes and birds) and poikilotherms. Challengingly, the CYP2H bird forms appear not to be monophyletic, so separating the mammalian CYP2CE from CYP2H is not possible (unless that side branch is the CYP2AF subfamily: see Almeida et al *Genome Biology and Evolution*, Volume 8, Issue 4, April 2016, Pages 1115-1131, <https://doi.org/10.1093/gbe/evw041>)? Amphibian CYP2s are said to be included - are they distributed across the tree (*Xenopus* has CYP2AC and others, but I am not certain how they're distributed).
- (e) Most recent data suggests that there were temperature maxima of 35-38C average 425Mya and again 100Mya. If Tm's are indeed 10-15C above optimal temperatures, then why is there a steady decrease to modern enzymes, rather than a more varied pattern? In addition, the steady decline from a high temperature origin is not readily reconciling the 'snowball earth' hypothesis, or even the better supported glaciation events (e.g Marinoan) that occurred 650-630 Mya.

That all said, however, there isn't data to support or refute the evolutionary arguments, and this is a very important contribution to the literature on enzyme stability and promiscuity.

I think supplemental Figure 10 could be included in the main figures, despite not showing a clear trend, as it's an interesting result and contributes to the stability/promiscuity trade off argument.

Minor note:

Supplemental figure 10C and D are upside down in the supplementary PDF.

Referee #3:

Review to:

Thermostable ancestors enabled evolutionary diversification of promiscuous chemical defence enzymes

Summary

The authors studied the evolution of CYP2, a p450 multifunctional enzyme family involved in chemical detoxification in vertebrates. The enzymes are multifunctional and known to have evolved through a significant expansion 500 million years ago when the Earth was no longer hot. The authors reconstructed 12 the ancestral states of the protein family. They measured their thermostability and catalytic activity against a battery of substrates. Then, they analyze the structure to understand how thermostability changes over time and how it relates to multifunctionality. The experimental designs aim to prove the hypothesis that multifunctional enzymes are kept thermophilic (robust) to allow their diversification.

The T_m of the 12 ancestors, plus extant enzymes, shows that the young ancestors are less thermostable than the old ancestors. This incremental reduction in T_m was then analyzed in terms of the enzymes' hydrophobic/hydrophilic interactions. They find that the highest thermostability is linked to better fold packing. The authors measure the enzyme's capacity to modify five different substrates and bind 48 chemically diverse ligands. They found that binding promiscuity and the age of the protein do not correlate. However, the most thermostable ancestor birthed the most functionally diverse clades.

Novelty and significance

The research is novel in that it adds to the discussion of the protein community related to "why ancestral proteins are thermostable, even when they are from times when the Earth was not as hot".

The fact that the authors show that is likely not a consensus approach but a lack of pressure to lose the thermostability of older proteins is of value.

Comments

1:

Ancestral sequence reconstruction (ASR) generates more thermostability than the extant enzymes from where they are inferred. In cases where the ancestors existed on Earth in times when the temperature was indeed high, this makes sense. However, the ASR of enzymes that evolved more recently, when the Earth was no longer hot, also leads to thermostable ancestral problems.

In the introduction, the authors mention two possible explanations: (1) The evolution from thermostable ancestors does not always lead to mesophilic enzymes. Drift can keep them thermostable. (2) Most enzyme families studied by ASR are multifunctional. Hence, they are supposed to stay robust (as in a proxy to thermophilic?) to allow for functional diversification.

Here, the authors should add a third explanation. While constructing ancestral sequences, there could be a bias towards consensus -which leads to thermophilic proteins. They only mention it later in the discussion.

2:

Towards the end of the introduction, the authors mention that thermophilic enzymes are supposed to be less multifunctional, as they should be less flexible. This contradicts the hypothesis they want to test, as thermostability should be "penalized" through the evolution of multifunctional enzymes. Perhaps a better explanation here is needed.

3:

Figures 4 and 5 could be merged into 1. Treating all substates equally regardless of whether they are endogenous or not. Also, some graphs are empty, or the activity can't be easily observed. When the enzymes are inactive, maybe don't add a graph, and for those with very little activity, try to change the Y-axis scale.

4:

A concluding remark at the end of the results section titled: "Catalytic activity of ancestral forms towards marker substrates" is missing. This was done for all other sections in the results, and I think it helped throughout the reading.

5:

Discussion, lines 481 - 495: The authors mention that compared to the oldest ancestor, CYP2ABGSFTCEH, newer proteins are more thermostable by 10 degrees. They assume drift increased thermostability. I wonder how significant these 10 degrees are? Maybe some statistical analysis is needed?

6:

Discussion, lines 495-500: Why is the lineal decrease in thermostability (Figure 2) proof that thermostability was lost by drift? It was not clear to me.

7:

Discussion, lines 511-518:

This part is not clear to me. Do the authors mean that organisms today growing in hot environments (thermophiles) have a proteome with a T_m 10-20 degrees Celsius higher than mesophilic? How is that related to the following statement?

7:

From the results section, it was hard to visualize the trend the authors refer to in the discussion (lines 571-572), namely, that the ancestors show broader substrate specificity. It seems that they are multifunctional, but also some extant enzymes. So, I'm not sure we can visualize the reduction of multifunctionality between ancestral and extant enzymes with this data.

8:

Discussion, lines 583-593.

The authors now contradict themselves in the data analysis. Before (point 7), it seems that the ancestral proteins were more promiscuous, but now they write that the ability to bind and monooxygenase compounds is similar between ancestral and extant proteins. But the catalytic rate was lower. A better explanation is needed.

Response to Reviews:**Referee #1:**

While I find this idea generally interesting, the paper makes some questionable methodological choices (see below). I am also not sure that the data presented here is sufficient to establish thermostability as a cause of diversification. In effect, the authors argue that substitution rates later in the tree are to a very large degree caused by the thermostability of ancestors. The statistical evidence for this claim is weak (indeed, no formal analysis is carried out at all to substantiate this), so in my view, this conclusion is speculation, which the authors are, of course, entitled to.

We agree with the reviewer that this is speculation; we would need to assess the relative thermostability and degree of diversification in a much larger set of protein families to prove this relationship. We say it is consistent with, or supports, the hypothesis rather than that it proves the hypothesis that thermostable ancestors enable more diversification. In the abstract we state:

“The major lineages differed in their relative diversification, with the more stable lineage leading to greater extant sequence diversity.”

(this is an observation, no more or less), and:

“This work supports the hypothesis that robust ancestors facilitate evolutionary diversification”.

In the discussion we say:

“The exceptional stability of the basal CYP2 forms *may* have permitted this extensive degree of diversification, which is *consistent with* the idea that high stability contributes to the evolvability of a protein (Bloom, et al. 2006; Besenmatter, et al. 2007) by allowing destabilizing but functionally useful mutations to be tolerated without compromising the protein fold.”

and

“This extra stability *may* be the reason the CYP2C lineage has given rise to roughly four times the number of functional extant forms than seen in other CYP2 subfamilies (Figure 1).”

In the Introduction we say:

“We chose the CYP2 family of P450s to test the hypothesis that a robust ancestor is necessary for rapid and extensive diversification and that in the absence of a selection pressure, such thermostability is lost incrementally over time.”.

However, we take the reviewers point that a reader may interpret our statements more definitively. In retrospect, this is not a hypothesis that can be proven or disproven with

this study, so we have altered this sentence (p. 6, now lines 117-118, previously lines 112-113) to:

“We chose the CYP2 family of P450s to explore the relationship between ancestral robustness and evolutionary diversification.”

Likewise, in the discussion, we have changed the following sentence (p. 24, lines 542-544, previously p.22, lines 477-478):

“We sought to test this hypothesis using a clade of the highly diverse, xenobiotic-metabolising CYP2 family as a model system.”

to:

“We sought to explore the change in thermostability across evolutionary time using a clade of the highly diverse, xenobiotic-metabolising CYP2 family as a model system.”

We have also changed the wording of the first sentence of the last paragraph of the discussion on p. 33, lines 770-771 (previously p.29, lines 665-666) from:

“In summary, this study has provided evidence to support the hypothesis that robust protein folds facilitate evolutionary diversification”

to:

In summary, the results of this study are consistent with the hypothesis that robust protein folds facilitate evolutionary diversification ...”.

As this paper is concerned with the thermostability of ancestral proteins, it has to deal with the debate around whether these very high stabilities (in some cases 70 degrees Celsius and above) are realistic. Such high melting temperatures are now reserved for proteins from thermophilic organisms that live in niches much hotter than what we can reasonably assume ancestral vertebrates to have experienced in the ocean. One particularly striking example is ancestor 2C, which, as far as I can tell, existed in an ancestral mammal - presumably an endothermic organism that regulated its body temperature. This protein is reconstructed to have a melting point of over 60 degrees Celsius. This puts it far outside the distribution of melting points in vertebrates (according to this recent paper <https://www.nature.com/articles/s41592-020-0801-4>).

Figure 2 of this recent paper (which we had cited) makes the point that the melting temperatures of proteins in different organisms are distributed across a range of temperatures spanning ~ 15-30 °C in total, with the breadth of the distribution depending on the organism. Indeed, even for *Homo sapiens* (for which we are very certain of the ambient and optimal growth temperature) there is a significant proportion of proteins that have T_m values above 60 °C (the peak of the distribution

being centred on ~ 53 °C). Thus, even in extant mammals which regulate their body temperatures and live in cooler times than existed when the first mammals arose ~ 225 Ma, there are proteins with similar T_{50} values to the CYP2C ancestor.

The authors argue that such stabilities are a remnant of a time when surface temperatures were even higher, and that stability only falls very slowly due to drift. I find this argument unconvincing. First, it is unclear to me from what time prior to the Cambrian the authors even believe these proteins inherited their extreme stabilities - could this perhaps be stated explicitly?

P450s are found in all domains of life so could have been present in the last universal common ancestor (LUCA). However, they appear to have been transferred horizontally to archaea (Ngcobo, et al. 2023) so probably first appeared in an ancient bacterium around the time of the Great Oxidation Event ~ 2.4 Ga and necessarily have survived and evolved through global temperature changes ever since. Estimates put the temperature of the Earth in the Archean era at 55-85 °C with an overall steady decline thereafter, notwithstanding fluctuations (e.g., two hothouse periods noted below) due to changes in geological activity and atmospheric chemistry. Temperatures around 2-2.4 Ga are estimated to have been in excess of ~ 60 °C (Robert and Chaussidon 2006).

A very recent analysis of average global temperatures during the Phanerozoic Era (Judd, et al. 2024) suggests that over the last 485 million years, the global mean temperature has varied between 11 and 36 °C but between 22 and 42 °C in the tropics. Based on their analysis, the authors explicitly stated that ‘ancient life must have evolved to endure extreme heat’. The peak hothouse temperature is proposed to have occurred ~ 89 -94 Ma whereas the most recent hothouse period was ~ 55.8 Ma during the Paleocene–Eocene Thermal Maximum, with a steady decline in temperature thereafter.

We have added a detailed discussion of these times and global temperature estimates in the discussion on p.25, lines 570- p. 27, line 616 (previously p. 23 lines 502- 525) as follows:

“These trends are consistent with thermostability being an evolutionary relic from enzymes that existed in very hot environments in primordial times, that was lost through genetic drift in the absence of selection pressure to maintain it. P450s are found in all domains of life so could have been present in the last universal common ancestor (LUCA). However they appear to have been transferred horizontally to archaea (Ngcobo, et al. 2023) so probably first appeared in an ancient bacterium around the time of the Great Oxidation Event ~ 2.4 billion years ago (Ga). Estimates put the temperature of the Earth in the Archean era at 55-85 °C with a decline thereafter. Temperatures around 2-2.4 Ga are estimated to have been in excess of ~ 60 °C (Robert and Chaussidon 2006). The CYP2 clade studied here dates from the earliest tetrapods, i.e., ~ 400 -420 Ma (George and Blieck 2011), meaning that the oldest ancestor

resurrected here corresponds to vertebrates that may have inhabited marine environments with average global temperatures of ~ 15-25 °C (compared to ~ 15 °C today) (De Vleeschouwer, et al. 2024) with temperatures up to ~ 40 °C in low latitudes (Grossman and Joachimski 2022). A very recent analysis of average global temperatures during the Phanerozoic Era (Judd, et al. 2024) suggests that over the last 485 million years, the global mean temperature has varied between 11 and 36 °C but 22 to 42 °C in the tropics. The peak hothouse temperature is proposed to have occurred ~ 89-94 Ma and the most recent ~ 55.8 Ma during the Paleocene–Eocene Thermal Maximum, with a steady decline in temperature thereafter.

“It is impossible to accurately evaluate whether such temperatures would have imposed any significant selection pressure on the maintenance of thermostability in proteins in putative ancestral vertebrates without knowing which latitudes they inhabited, and their ability to regulate their temperature by either physiological (homeothermic) or behavioural means (e.g., moving to cooler environments). Nor is it possible to define the time at which a particular putative ancestor would have existed except within very broad windows of evolutionary time. For example, the CYP2C and CYP2E subfamilies are both present in marsupials and eutherian mammals, so logically must have been in their most recent common ancestor (MRCA). They both derive from the CYP2CE ancestor which covers (only) the same clades. Logically, the CYP2CE must have predated the CYP2C and CYP2E ancestors, but we have no way of ascertaining when the duplication occurred. It must have occurred between the MRCA of the eutherian mammals and marsupials at ~ 160 MYA and the MRCA of the mammals and sauropsids (reptiles and birds) at ~ 319 MYA but when exactly – or even approximately – in this ~159 MY period is unclear.

Bearing in mind these caveats, it is notable that the average melting temperature of the proteome was ~ 10-20 °C higher than the respective optimal growth temperature (OGT) across a range of organisms exhibiting different OGTs (Jarzab, et al. 2020). The distribution of melting temperatures ranged ~ 10-20 °C above and below the average for the three vertebrates analysed. Thus, average melting temperatures of 50-60 °C would not be unreasonable for proteins in organisms surviving hothouse environments of ~ 30-40 °C and the range could extend up to 80 °C. We cannot assign precise ages to any of the hypothetical ancestral proteins resurrected here, but the deeper ancestors (i.e., CYP2ABGSFTCEH, CYP2ABGSFT, CYP2ABGS, CYP2CEH, CYP2CE and CYP2BS, that could have existed closer to times of hothouse temperatures than the single subfamily ancestors) showed t_{50} values from ~ 59 – 83 °C, placing them towards the top of or above the expected range.”

But more importantly, this theory seems at odds with overwhelming biochemical evidence that most mutations are destabilizing. In the absence of continuous selection for stability, proteins should therefore evolve towards the minimal stability they can function with relatively quickly (I am sure the authors are aware of Richard Goldstein's arguments). At the very least, the authors have to wrestle with this inconsistency in their theory. As it stands, the argument feels like motivated reasoning to explain away otherwise extremely puzzling reconstructed stabilities.

We thank the reviewer for pointing out Goldstein's studies which we agree are very relevant. We have taken to opportunity to discuss his theory in the revised manuscript. From simulations of evolution under specific assumptions, Goldstein has hypothesized that proteins will evolve to be only as stable as they need to be to fulfill their functional role. This theory is consistent with the idea that thermostability is lost by genetic drift over time, but would imply that a close temporal relationship should be seen between global temperature shifts and the changes in thermostability observed in proteins. However, it is not clear how the assumptions used for these simulations relate to natural evolution of a protein family like the CYP2s.

Importantly, the functional role of enzymes in this family is dependent upon the ability to evolve rapidly in response to animal-plant chemical warfare, so there may be selection pressure to maintain stability and thereby mutational robustness. It is straightforward to imagine bottlenecks where having a high degree of polymorphism in an enzyme used for detoxification (i.e., the ability to rapidly evolve function in a chemical defence gene) might have conferred a survival advantage. Selection pressure from dietary toxins may be mild under most conditions but potentially very strong e.g., after a famine where foraging is restricted to specific plants containing toxic secondary metabolites.

We have also discussed the *in vitro* evolution studies on P450s by the Arnold lab which predict the opposite effect, i.e., that evolution will favour the development of excess mutational robustness, i.e., enhanced protein stability, where there is a sufficiently large effective population and a high degree of polymorphism (both of which could be argued pertain here).

The following text has been added to the discussion (p. 27, line 632 - p. 28, line 658):

“Evolutionary simulations have suggested that proteins evolve quickly to a point where they are marginally stable (Goldstein 2011), which is consistent with the idea that thermostability is lost by genetic drift, but would imply a close temporal relationship should be seen between global temperature shifts and the changes in thermostability observed in proteins (assuming generation times are short relative to the timescale of climate changes). It is not clear how the assumptions used for these simulations relate to natural evolution of a protein family like the CYP2s.

In contrast, prospective *in vitro* evolution studies on a bacterial P450 have revealed that neutral evolution leads to excess mutational robustness via enhancement of protein stability in populations of a sufficient size and diversity (Bloom, et al. 2007). While this effect would be expected to be more significant in microbial populations than vertebrates (Bloom, et al. 2007), given the scope of the CYP2 family under analysis here, and the degree of polymorphism evident in extant enzymes (Gaedigk, et al. 2021), the effective population of P450s under consideration here may be sufficiently large and diverse to fit these criteria (or may have been at some point in the past), leading to elevated protein stability and enhanced tolerance of mutations, explaining in part the maintenance of thermostability at mesophilic temperatures.

Importantly, thermostability does not only confer resistance to elevated temperature (Trudeau, et al. 2016). The functional role of most P450s in the tetrapod CYP2 clade analysed here is dependent upon the ability to evolve in response to animal vs. plant chemical warfare, so there may be selection pressure to maintain stability and thereby mutational robustness. It is straightforward to imagine bottlenecks where having a high degree of polymorphism in an enzyme used for detoxification (i.e., the ability to rapidly evolve function in a chemical defence gene) might have conferred a survival advantage. Selection pressure from dietary toxins may be mild under most conditions but potentially very strong e.g., after a famine where foraging is restricted to specific plants containing toxic secondary metabolites.”

I want to be clear here that I know that the senior author's group stands on one side of this argument about the stability of ancestral proteins and this reviewer stands on the other. I have no intention to grandstand in this review process, but I think the paper deserves a frank discussion of the plausibility of these extremely high stabilities. As I explain below, the paper would have benefited from quantifying reconstruction uncertainty.

We have added text to the discussion as detailed in response to other comments that addresses the plausibility of the stabilities we measure, including a consideration of whether they could be artefactual due to a ‘consensus effect’ (page 28, line 659 - p. 29, line 673) and how they compare to the ‘meltome’ analysis of Jarzab et al., 2020. We also address the confidence in the ancestral sequence reconstruction by including a tree with bootstrap values and assessing the statistical significance of changes in T_m between ancestors (Appendix Table S3). In the context of the Jarzab et al. (2020) ‘meltome’ analysis and ASR studies on other proteins over a similar evolutionary period (Trudeau, et al. 2016; Garcia, et al. 2017), which have shown similar effects, we believe the thermostability we have observed for P450s is plausible. (We have introduced additional citations to these prior studies in the Discussion on p. 23 at line 526 and p. 24, line 540). Nevertheless, there are relatively few comparator studies and

the field would benefit from more research on how stability changes over evolutionary time for proteins from different families and diverse functional roles as we note now at the end of the discussion on p. 33, line 783 - p. 34, line 789. We hope our study will prompt such research.

Apart from these conceptual issues, I have a few methodological questions.

I am puzzled by the unjustified use of joint reconstructions. This is not standard in the field. The reason is that in a joint reconstruction, errors in the reconstructions deep in the tree propagate through all nodes in the tree. In a marginal reconstruction, each node is reconstructed whilst integrating over the uncertainty at all other nodes, meaning that such propagation of reconstruction errors is much less of a problem. Why did the authors choose a joint reconstruction?

A study such as this that involves a considerable amount of biochemical characterization of multiple ancestral and extant forms inevitably requires an extended period to complete. The ancestral sequence reconstruction (ASR) was undertaken at the very start of the study, so there was necessarily an extended period between performing the ASR and submitting the study for publication. When we started this work, both marginal and joint reconstruction were considered valid approaches for ASR. We chose joint since we explicitly wanted to compare multiple nodes across a tree; therefore, all were taken from the same statistical distribution to ensure consistency across ancestors.

It is not clear from a review of the literature whether most studies favour marginal or joint maximum likelihood reconstruction, i.e. which is 'standard'. Many papers fail to specify the choice but only cite the tool used, which is ambiguous when many tools implement both alternatives. Notwithstanding this lack of information, the orthodoxy that has been established by specific groups appears to favour marginal for resurrection of a specific ancestor. Marginal also has the useful benefit of generating posterior probabilities. Nevertheless, prominent peer-reviewed publications (e.g. Konno et al in *Commun Biol* last year (Konno, et al. 2024), our 2018 paper in *Nature Catalysis* and our 2022 paper in *Mol. Biol. Evol.*) have all reported joint reconstructions and been published in stringent, high impact factor journals.

To explore the variation between joint and marginal inferences, we performed a marginal reconstruction using the same alignment and tree. With one exception (at 96.36% for CYP2BS, a poorly supported node in the tree), the ancestors obtained were at least ~98% identical to the respective joint ancestor at the same position (Appendix Table S2). Over the full ~500 residue open reading frame of the expressed proteins, this typically equated to 11 or fewer differences (16 in the case of CYP2BS). When the marginal and joint ancestors were directly compared (Appendix Table S2), most discrepancies were seen to involve conservative substitutions (e.g., of one hydrophobic residue for another). Notwithstanding the possibility that a single substitution in a critical position could change the activity of a P450, our previous

experience making a library of proteins that differed at 10 positions (Gumulya, et al. 2018) showed quantitative changes in activities but not frank qualitative differences across the library.

These comparisons have now been described in the Results section (p. 16 line 358 – p.17, line 363) with reference to the new Appendix Table S2. The paragraph previously in the Discussion starting with “The proteins resurrected here represent the most probable ancestors...” has been relocated to the beginning of the Discussion (p.22, line 498 - p. 23 line 522) where the following text has been introduced:

“The proteins resurrected here represent the most probable ancestors given the available sequence information, evolutionary model and algorithms chosen for ancestral inference. Importantly, they do not necessarily reflect any one historically accurate ancestral enzyme but rather represent the most likely candidate of a set of possible sequences. As such, there is a degree of uncertainty in any observed structural and functional features as with any ancestral sequence reconstruction. Joint reconstruction was selected here, since the aim was to compare ancestors across a consistent reconstruction, and the GRASP tool (Foley, et al. 2022) was chosen to maximize the amount of sequence information that could be used. A comparison of the ancestors predicted by joint and marginal maximum likelihood approaches, showed little variation (generally less than ~2%) in the sequence of the inferred ancestors (Appendix Table S2). Likewise, the variation in sequences caused by accommodating variation in substitution rates was < 2% (Appendix Table S7). The greatest effect observed on ancestor sequences was seen with the choice of tree (Appendix Table S7), as found by others (Muñiz-Trejo, et al. 2025). The choice of tree was informed by the expected species relationships and a previously published phylogenetic analysis (Kirischian, et al. 2011) and statistical support was low for some branch points. However, when repeated with a different tree generated in IQTree that ignored established phylogenetic relationships, the bootstrap values were still low at some nodes. The ancestors at comparable nodes from different trees were generally > 90% identical (all were at least 85% identical; Appendix Table S7), and the positions that were inferred differently corresponded to those showing lower posterior probability in the marginal reconstruction. Importantly, the tetrapod CYP2C sequence clade selected for reconstruction is relatively rich in sequence information from heavily studied higher vertebrates, with representation across different animal classes and at key evolutionary branchpoints, all of which are conducive to greater accuracy in the ancestral inference (Foley, et al. 2022), since they provide more information on which to build a more accurate tree.”

In addition, the inference of the marginal ancestors and their posterior probabilities has been described in the methods on p. 10, at lines 192-197:

“For comparative purposes, marginal reconstructions were inferred via GRASP at each node of the same tree allowing calculation of the posterior probabilities of all residues at each position of the alignment and derivation of an ‘AltAll’ ancestor, in which the second most probable residue was substituted at any position of the marginal sequence where the second most probable residue had a posterior probability of 0.4 or greater.”

In addition, the model is incompletely specified. The substitution matrix is JTT, but what about the state frequencies? Is rate heterogeneity accounted for using a gamma distribution? How was this model chosen? A quick glance at the T-REX web interface seems to suggest that no model finding routine is implemented there.

At the time the reconstruction was performed, model finding algorithms were not routinely available and JTT, WAG and LG were commonly implemented. A previous study showed little effect of changing the model (Abadi, et al. 2019), and JTT was an established choice in the field at the time, so it was used.

A recent study from the Thornton lab showed little effect of rate heterogeneity and the specific model used on the inference (Muñiz-Trejo, et al. 2025). The primary determinant is phylogenetic signal (i.e., tree topology) meaning that it is better to base the ASR on more sequences than address rates. This was the basic reason we used GRASP - since it could sample more sequences than other tools available at the time.

Re-running the ASR using GRASP with a rate file implemented using the same alignment and tree gives ancestors that are 98-100% identical (Appendix Table S7). Redoing the joint ASR with the same alignment but a new tree without a rate file gives ancestors that are ~90-96% identical (Appendix Table S7).

The same goes for the reconstruction with GRASP. This program's chief advantage is a somewhat better way to assign gaps. It comes with the very considerable disadvantage of only being able to incorporate rate heterogeneity when the alpha parameter of a gamma distribution is externally supplied. The authors do not state whether this was done here, so I suspect that it wasn't. At the very least, the use of GRASP in this case would have to be carefully justified, as a much more appropriate model of evolution (i.e., one that models rate heterogeneity among sites) was traded in for an uncertain advantage in automated gap assignments.

The principal advantage of GRASP, especially at the time the reconstruction was performed, was that it was able to handle many more sequences than other algorithms available at the time. Given we were dealing with a very large and diverse protein family, we wanted as much information as possible on which to base the inference. The improved gap handling noted by the reviewer is an additional benefit when dealing with such large and diversified family such as this, where indels feature commonly, in contrast to reconstructions of more conserved proteins for the structure is more constrained by the need to efficiently fulfill defined physiological functions.

At the stage the inference was performed, GRASP did not model rate heterogeneity. Our priority was to sample as much diversity as possible, i.e. maximise the number of sequences across the phylogeny. A later study has shown the value of that choice in that topology has more influence than rate heterogeneity (Muñiz-Trejo, et al. 2025).

We have added a comment in the first paragraph of the discussion (p. 22 lines 503-505) to explain the decision to use GRASP:

“Joint reconstruction was selected here, since the aim was to compare ancestors across a consistent reconstruction, and the GRASP tool (Foley, et al. 2022) was chosen to maximize the amount of sequence information that could be used”.

We have also included the comparison of the sequences reconstructed with and without a gamma distribution in the deposited data file supporting Appendix Table S7: AncNodes_diff_trees_with_vs_wo_RF.fasta.

I could find no information about the quality of the reconstructions in terms of their posterior probabilities. This information has to be provided, and the tree, alignments, and reconstructions need to be deposited somewhere public.

Posterior probabilities are not generated via the joint reconstruction method and therefore were not supplied. However posterior probabilities were obtained for the corresponding marginal reconstruction (deposited supporting data file: Posterior probabilities Marginal_&_Joint_Inferences.xlsx). In addition, the posterior probabilities of the joint ancestors that were resurrected have been extracted from those obtained via the marginal reconstruction, and are included in the same tables. A comment has been added to the methods on p. 10, at lines 192-197 describing the inference of the marginal ancestors as noted above.

All supporting source data has been uploaded according to the EMBO J instructions.

Similarly, it appears that the authors have not tested for the effect of statistical (or indeed topological) uncertainty on the exact sequences and properties of their ancestors. This is very disappointing and not up to the standard of the field. Especially for quantitative parameters like stability, this should absolutely be done.

The statistics for the thermostability data have been added to Appendix Table S3. Since it would have made Figure 2 too congested to show indicators of statistical significance there as well, we have instead shown the independent replicates on the figure rather than means +/- SD, so the reader can evaluate the experimental error in the data directly. We have also added the following comment to the legend:

“All ancestors showed significantly different $^{60}T_{50}$ values to their immediate ancestor along the same lineage except CYP2ABGSFT (compared to CYP2ABGSFTCEH) and CYP2ABGS (compared to CYP2ABGSFT); two tailed, heteroscedastic Student's *t*-test. All extant enzymes showed $^{60}T_{50}$ values

that were significantly lower than all their cognate ancestors ($p < 0.006$, two-tailed heteroscedastic Student's t-test).”

Statistics have also been added to comments on pp. 18 (lines 387-388), 19 (lines 418-419), 24 (lines 550-551), 25 (line 569) and 27 (lines 627-628) where thermostabilities are compared. The full matrix of p values is presented in the deposited supplemental data file: Thermostability & expression data from Supp Table 3 with stats.xlsx.

Statistical analysis had already been performed and shown for the turnover of testosterone, the alkoxyresorufins and the luciferin analogues (but statistics are only shown on the supplemental figures for clarity, not figures 4 and 5 of the main text). Significance values have now been added to Appendix Figure S4 (coumarin turnover; formerly Appendix Figure 5).

A single value is obtained in the Tanimoto distance analysis for each ancestor, and we do not suggest any correlation in Figure EV3 (formerly Supplementary figure 10).

No comparisons are being made using the data in Appendix Table S6; means +/- standard deviations are simply provided where a spectral binding interaction was observed.

I could not clearly assess the plausibility of the trees, as the trees are not labeled in a way that allows the reader to compare them to known species relationships and whole genome duplications in the vertebrates. There are no branch supports on the trees, and a phylogeny with all taxa labeled is now available as Figure EV1.

We had noted bootstrapping had been done in the Methods but not shown the values. An additional tree with bootstrap values and all taxa labelled has now been included as Figure EV1. This tree represents the one inferred that showed best agreement with the expected species tree for the largest number of clades. In addition, we were mindful of the established phylogeny for the CYP2 family in the literature (Kirischian, et al. 2011). For example, the CYP2BS node did not have strong bootstrap support but was predicted from the Kirischian et al., 2011 phylogeny. We chose to use the maximum likelihood tree that best reflected the expected species relationships and established phylogeny rather than use a tree with better bootstrap support and subjectively rearrange clades to match the expected relationships.

I also would strongly discourage the authors from using the circular representation of their tree. This representation makes the branching relationships very hard to see and makes it impossible to compare the relative ages of the nodes in terms of branch length. All this representation does is to emphasize that the authors sampled more sequences from the 2C clade than from any other clade.

Figure 1 has been converted to rectilinear format and major phylogenetic groupings have been annotated (the detailed phylogenetic groupings being available in the very much larger tree in the Figure EV1).

I don't understand why the authors chose to use relative branch lengths as a measure of age. The divergence times of most vertebrate groups have been estimated many times over and are freely available on TimeTree. On a vertebrate protein phylogeny, it should be simple enough to assign each node to either a speciation node on the species tree, or an interval between speciation nodes for nodes that correspond to gene duplications on the gene tree. Using branch lengths as a proxy for age tacitly assumes that these genes evolve in a clock-like manner, which I highly doubt.

This is more complicated than it would appear initially, because we are not simply tracking the evolution of an enzyme that diverges simply due to speciation. If the gene tree matched the species tree exactly, we could assign ages from Timetree to each of the nodes. However, in many cases, ancestors that are next to each other are present in exactly the same group of animals. For example, both CYP2C and CYP2E are present in marsupials and eutherian mammals, so logically must have been in their most recent common ancestor (MRCA, the stage for which Timetree provides estimates of age). They both derive from the CYP2CE ancestor which covers (only) the same clades. Logically, CYP2CE must have predated CYP2C and CYP2E, but how should we represent its age relative to those of its descendants: we have no way of ascertaining when the duplication occurred. It must be sometime between the MRCA of the eutherian mammals and marsupials at ~ 160 MYA and the MRCA of the mammals and sauropsids (reptiles and birds) at ~ 319 MYA but when exactly – or even approximately – in this ~159 MY period? Likewise, gene losses complicate the interpretation: for example, there is no evidence of a CYP2C-like gene in monotremes, suggesting it has been lost in that lineage, yet there are CYP2C clades in reptiles and birds (branching earlier than the ancestor we have defined as CYP2CE). The only indicator we have of the evolutionary distance between the CYP2CEH and CYP2ABGSFTCEH is the branch length. However, as the reviewers have pointed out, evolutionary rates can vary and so branch lengths cannot be taken as a surrogate of age.

The reviewer suggests using either a speciation node or “an interval between speciation nodes for nodes that correspond to gene duplications on the gene tree”. In this case we could assign CYP2C and CYP2E ancestors to a speciation node and CYP2CE to an interval of ~ 159 MY. CYP2CE and CYP2ABGSFTCEH would necessarily be consigned to the same interval of between the MCRA of the amniotes (reptiles, birds, mammals) at 319 MYA and that of the tetrapods (amniotes plus amphibians) at 352 MYA (33 MY *in toto*). Given the width of these intervals, we do not believe this would produce a meaningful graph.

We have assessed different alternative scenarios to represent these data but all involve some assumptions about exactly when duplication events occurred, which compromise the validity of the comparison being made. Therefore, we have rephrased the comments regarding ‘evolutionary age’ to ‘evolutionary distance from the oldest ancestor’ (e.g. p. 17 lines 374 and 383-386; p. 24, line 552). We can comment on the order of gains or losses in thermostability, i.e., that older ancestors

are in general more thermostable. However, the rate at which these changes occur cannot be determined. The value of the figure, therefore, is in showing the general loss of thermostability with respect to the extent of divergence, which is directly related to branch length, i.e. the number of mutations between different ancestors.

We have removed the comments in the legend to Figure 2 and the last paragraph of the results (previously p. 20, lines 448-449) that suggested branch length was used as a surrogate for evolutionary age. We have also added a comment to the results on pp. 17, lines 384-386 to strengthen the caveat that branch length does not scale to evolutionary age due to the likelihood of variation in rates of evolution:

“Branch lengths are used to indicate the evolutionary distance between different ancestors but cannot be used as a surrogate for historical time due to the likelihood that amino acid substitution rates vary across time, species and position in the protein structure.”

We have also added text to the discussion on p. 26, lines 596-605 to explain the difficulty in assigning ancestors to specific times:

“Nor is it possible to define the time at which a particular putative ancestor would have existed except within very broad windows of evolutionary time. For example, the CYP2C and CYP2E subfamilies are both present in marsupials and eutherian mammals, so logically must have been in their most recent common ancestor (MRCA). They both derive from the CYP2CE ancestor which covers (only) the same clades. Logically, the CYP2CE must have predated the CYP2C and CYP2E ancestors, but we have no way of ascertaining when the duplication occurred. It must have occurred between the MRCA of the eutherian mammals and marsupials at ~ 160 MYA and the MRCA of the mammals and sauropsids (reptiles and birds) at ~ 319 MYA but when exactly – or even approximately – in this ~159 MY period is unclear.

In addition, the whole justification for the plausibility of these extremely high melting temperatures is that these proteins are thought to exist at times when the ocean temperature was high. It therefore seems essential to me that the reader can assess that these nodes are really that old. It would also help to plot ocean temperatures as a function of time somewhere in this manuscript.

We agree that it would be helpful for the reader to see directly how ocean temperatures are thought to have varied across geological time, but do not believe it is appropriate to insert a graph of ocean temperatures from other papers. Moreover, although there is broad consensus between studies, the estimates from various studies differ in the finer detail. Instead, we refer the reader to the excellent previous studies of Robert and Chaussidon (Robert and Chaussidon 2006), Judd et al., (Judd, et al. 2024), De Vleeshouwer et al. (De Vleeschouwer, et al. 2024) and Grossman and Joachimski (Grossman and Joachimski 2022), each of which provides useful and complementary plots.

Instead, we have introduced text in the discussion on p. 25, lines 579-581, to note the salient points from these studies:

“Estimates put the temperature of the Earth in the Archean era at 55-85 °C with a decline thereafter. Temperatures around 2-2.4 Ga are estimated to have been in excess of ~ 60 °C (Robert and Chaussidon 2006).”

And on p. 26, lines 586-591:

“A very recent analysis of average global temperatures during the Phanerozoic Era (Judd, et al. 2024) suggests that over the last 485 million years, the global mean temperature has varied between 11 and 36 °C but 22 to 42 °C in the tropics. The peak hothouse temperature is proposed to have occurred ~ 89-94 Ma and the most recent ~ 55.8 Ma during the Paleocene–Eocene Thermal Maximum, with a steady decline in temperature thereafter. “

The reviewer writes that “the whole justification for the plausibility of these extremely high melting temperatures is that these proteins are thought to exist at times when the ocean temperature was high.” Rather, we propose a hypothesis with which our data are consistent, but which we cannot prove, that the stabilities are an evolutionary relic from primordial times that are being lost with diversification of the proteins. The rate at which this occurs may be slower than otherwise expected due to the need for this enzyme family to be able to mutate rapidly in response to environmental challenge from plant secondary metabolites, due to its role in chemical defence. We do not propose that global temperatures were sufficiently high to impose selection pressure to maintain or evolve very high melting temperatures in the ancestors characterised. Indeed we state the reasons why it is not possible to assess such potential selection pressure from global temperature changes on p. 26 lines 592-596:

“It is impossible to accurately evaluate whether such temperatures would have imposed any significant selection pressure on the maintenance of thermostability in proteins in putative ancestral vertebrates without knowing which latitudes they inhabited, and their ability to regulate their temperature by either physiological (homeothermic) or behavioural means (e.g., moving to cooler environments).”

However, we do consider what has been documented of the relationship between average proteome melting temperature and optimal growth temperature in the context of the stabilities we observed. We have strengthened the text to try to clarify these finer points. On p.26, lines 606 (previously p. 23, lines 518-519) we reinforce the caveats by changing:

“However, it is intriguing...”

to:

“Bearing in mind these caveats, it is notable...”

We have amended the last paragraph of the discussion to clarify the points we make as follows (p. 33, line 777 - p. 34, line 790):

“The thermostability of the CYP2 tetrapod clade appears to have been lost in parallel with the general cooling of global temperatures since ~ 500 Ma (Ontiveros, et al. 2023). Observed stabilities are towards the upper extreme of, or exceed, those expected for vertebrates throughout this period, even accounting for fluctuations in global temperature and latitudinal differences between ecosystems. However, this excess stability and the resulting increased tolerance of mutations may have been important for the rapid evolution of this family of detoxification enzymes in response to animal vs. plant chemical warfare. ASR of other vertebrate protein families with different physiological roles could reveal whether the trends in thermostability evident for P450s and in other isolated examples (Bar-Rogovsky, et al. 2013) are seen more generally in the evolution of other proteins across the same geological timeframe or whether they are a feature of enzymes that require a high degree of polymorphism to respond rapidly to constantly changing selection pressures such as dietary toxins. Such studies could also provide clues as to the environmental conditions experienced by early vertebrates (Garcia, et al. 2017; Ontiveros, et al. 2023).”

Figure 3 purports to report a correlation between hydrophobicity and thermostability. The analysis provides only a visual representation of this supposed correlation and no statistical analysis. If the authors want to prove this correlation, they need to carry out an appropriate phylogenetic statistical test (like phylogenetic independent contrasts)

We thank the reviewer for the suggestion of the appropriate statistical test to more accurately assess this apparent correlation (which we have tried to show more clearly by adding a second panel to Figure 3). We have since performed a phylogenetic independent contrast analysis and found the apparent correlation failed to reach statistical significance. Therefore, we have included an additional Appendix Table S4 that reports the statistical analysis. We have also amended the relevant paragraph in the results (p. 18, line 410 - p. 19, line 419) as follows:

“This analysis revealed that the most thermostable forms had a higher degree of hydrophobicity in their buried residues and a higher degree of hydrophilicity in their exposed residues when compared to the less stable forms (Figure 3). This trend was most apparent when looking at the whole protein. When the fold was divided into discrete structural regions the trend weakened, suggesting a cumulative effect across the entire sequence (Appendix Figure S2). We performed a phylogenetic independent contrast analysis (Garland, 1992) on the twelve extant sequences to identify any

correlation between the buried / exposed values and thermostability, while accounting for the phylogenetic structure. Buried residues showed a moderate correlation but without reaching statistical significance (p-value > 0.1) (Appendix Table S4).”

In the last paragraph of the discussion (p. 33, line 774-776; previously p. 29, lines 669-671) we have changed:

“... segregation of hydrophobic and hydrophilic residues in the core and surface of a protein is important in determining stability....”

To:

“... segregation of hydrophobic and hydrophilic residues in the core and surface of a protein may be important in determining stability....”

We have also added text in the abstract (p. 2, lines 34-35) to indicate that there is a weak trend towards more hydrophobic buried residues and more hydrophilic solvent-exposed residues in thermostable but that the correlation is not statistically significant. The failure to reach statistical significance is also noted explicitly in the Discussion (p. 30, lines 685-686). Details of the analysis have been added to the Methods (p. 12, lines 244-247).

This analysis prompted us to undertake a more detailed analysis of the residue interaction networks amongst the ancestors, focusing on the greatest differences in thermostability that occurred with the least change in sequence, namely among the three oldest ancestors CYP2ABGSFTCEH, CYP2CEH and CYP2ABGSFT. These forms were 95-98% identical (Appendix Table S1). This analysis is presented in deposited data file: RING Analysis.xlsx. CYP2ABGSFTCEH had 24 fewer interactions than its more thermostable descendant, CYP2CEH, but 14 more than CYP2ABGSFT, which was slightly but not significantly less stable (Appendix Table S3). CYP2CEH had 10 more interactions than CYP2ABGSFT. The great majority of interactions were conserved or conservatively substituted among at least two of the three ancestors, but CYP2CEH had 156, CYP2ABGSFTCEH had 115 and CYP2ABGSFT had 96 unique interactions (deposited data file: RING Analysis.xlsx), in line with their order of thermostability. No clear candidates could be identified as critical for gain or loss of thermostability, with changes distributed throughout the length of the sequences.

A paragraph has been added to the Methods (p. 12, lines 248-253) and Results (p. 19, lines 420-430) describing this analysis.

Referee #2:

Review of EMBOJ-2025-120404

Thermostable ancestors enabled evolutionary diversification of promiscuous chemical defense enzymes (Raine et al)

This superb and interesting paper characterizes the ancestrally reconstructed cytochrome P450 family 2 enzymes. These are xenobiotic (and endogenous compound) metabolizing enzymes with a broad range of substrate diversity. Multiple different CYP2 enzymes have evolved in vertebrates, and this research has focused on the more 'recent' tetrapod evolutionary set (that is, those enzymes that have diverged in the last 380 million years). This work is really fascinating - it touches on protein function and structure relationships, with important implications for industrial applications, as well as very interesting questions about diversity and stability in protein evolution, and the evolution of organisms across hundreds of millions of years.

The paper is very clearly and compellingly written, and I have not found any typographical errors. The methods are well described and justified. The ASR is performed with one of the state of the art methods (GRASP), that takes into account indels, although there are few in the collection. This is not a structural modelling paper, so the use of only 5 models for each homology model is fine. The conclusions are careful, and carefully justified. A few suggestions, and discussion points of note:

(a) the oldest ancestors were the least well expressed - are there any obvious amino acid composition differences? Do these ancestors have relative levels comparable to the CYP1 or CYP3 families, that have similarly been reconstructed?

An extensive comparison of the amino acid composition of the ancestors failed to reveal any associations of amino acid composition with expression yield, either across the entire sequence or specific locations. The expression yields obtained here are in the same overall range as seem with the CYP3 and CYP1 ancestors resurrected previously.

A comment to this effect has been added to the Results (p.17, lines 370-373):

“a range that was comparable to yields observed previously for ancestors resurrected from the CYP3 (Gumulya et al., 2018) and CYP1 families (Harris et al 2022)”.

(b) Could the plot in Figure 2 have approximate divergence ages as an additional x axis, e.g. based on Timetree?

As noted above in the response to a similar comment by Reviewer 1, This is not straightforward due to the inability to pin down ancestors to specific divergence points where they cover the same phylogenetic groups. (Please refer to the comment above.)

(c) One method of examining the range of ASR possibilities is to express the 'alt_all' sequence, that incorporates all the lower probability amino acids at ambiguously reconstructed positions (Eike et al Mol Biol Evol. 2016 Oct 30;34(2):247-261. doi: 10.1093/molbev/msw223). How does the alt_all ML reconstruction compare to the consensus reconstruction?

We have derived the AltAll sequences from the marginal reconstruction and compared them to the most probable ancestors from both the marginal and joint reconstruction as well as the consensus sequences as suggested by the reviewer. The AltAll corresponds to an alternative inference where the next most probable residue was substituted wherever it had a posterior probability of 0.4 or greater.

There were 8.0 +/- 3.6 (mean +/- SD) differences between the marginal and the AltAll (range, 3-15 differences) and 11.3 +/- 5.2 differences between the joint and the AltAll (range, 2-23 differences). This compares to 8.6 +/- 4.1 between the joint and marginal most probable ancestors (range, 2-18 differences). These data are included in a table of the average posterior probabilities of the joint vs marginal ancestors in the archived data (Excel file: "Ave Joint, Marg & AltAll Post Probabilities_rev.x").

Between each AltAll and its corresponding consensus sequence, the percent identity ranged from 70.16 to 95.26% (see the table below), i.e., over approximately the same range as between the Joint ancestors and consensus sequences and consistently lower than between the Joint, Margin and AltAll ancestors. This data has been deposited with the other comparisons involving the consensus sequences in the uploaded source data.

Consensus sequence	AltAll ancestor	% identity
2ABGSFTCEH_consensus	AltAll_N4	73.09
2CEH_consensus	AltAll_N5	70.16
2CE_consensus	AltAll_N8	71.65
2C_consensus	AltAll_N10	80.47
2E_consensus	AltAll_N357	85.32
2ABGSFT_consensus	AltAll_N559	76.2
2F_consensus	AltAll_N562	90.99
2ABGS_consensus	AltAll_N623	79.28
2BS_consensus	AltAll_N624	71.96
2B_consensus	AltAll_N625	80.97
2S_consensus	AltAll_N727	79.22
2A_consensus	AltAll_N851	95.26

Sequences were compared across the whole inferred sequence.

(d) This family reconstruction encompasses both homeotherms (evolving separately in amniotes and birds) and poikilotherms. Challengingly, the CYP2H bird forms appear not to be monophyletic, so separating the mammalian CYP2CE from CYP2H is not possible (unless that side branch is the CYP2AF subfamily: see Almeida et al Genome Biology and Evolution, Volume 8, Issue 4, April 2016, Pages 1115-1131, <https://doi.org/10.1093/gbe/evw041>)? Amphibian CYP2s are said to be included - are they distributed across the tree (Xenopus has CYP2AC and others, but I am not certain how they're distributed).

Additional subfamily names have now been added where relevant in the revised version of Figure 1. CYP2H is monophyletic: the neighbouring clades are CYP2AH, CYP2AR and the sauropsid CYP2C clade. The CYP2AF subfamily lies outside the clade under consideration. Several CYP2 subfamilies that are present in other tetrapods are absent from the birds, confounding direct comparisons of the tree presented here and that in Almeida et al (2016), since the phylogeny developed in that study only considered subfamilies found in birds.

(e) Most recent data suggests that there were temperature maxima of 35-38C average 425Mya and again 100Mya. If Tm's are indeed 10-15C above optimal temperatures, then why is there a steady decrease to modern enzymes, rather than a more varied pattern? In addition, the steady decline from a high temperature origin is not readily reconciling the 'snowball earth' hypothesis, or even the better supported glaciation events (e.g Marinoan) that occurred 650-630 Mya.

Our reconstruction does not go back deeper than the vertebrates, which arose ~ 580-600 Mya. Therefore, we cannot comment the pattern of changes in thermostability before then.

Regarding the shape of the decline to the modern ancestors, we do not propose that the thermostability we see parallels global temperatures exactly. There are many factors that could mitigate the effect of global temperatures as a selection pressure on protein stability as noted on p. 26, lines 592-596.

Importantly, we would expect more variation in the thermostability vs. branch length were we to resurrect more ancestors, simply due to stochastic changes, irrespective of global temperature variations. Importantly, we resurrected only twelve ancestors across two lineages. The nodes were selected to cover broad evolutionary distances as well as to represent the major xenobiotic-metabolising lineages. There is not sufficient fine detail in the ancestors chosen to reveal more than broad changes.

That all said, however, there isn't data to support or refute the evolutionary arguments, and this is a very important contribution to the literature on enzyme stability and promiscuity. I think supplemental Figure 10 could be included in the main figures, despite not showing a clear trend, as it's an interesting result and contributes to the stability/promiscuity trade off argument.

We are happy to move what was formerly Supplemental Figure 10 to the main text if the editor concurs but appreciate the pressure on space in the journal and this is a

quite large figure. Since there was no correlation observed and neither of the other reviewers or the editor mentioned this analysis, we have instead selected this figure for inclusion as an extended view figure (Figure EV3).

Minor note:

Supplemental figure 10C and D are upside down in the supplementary PDF.

We thank the reviewer for picking up this odd glitch that appears to have occurred in the conversion to pdf. The tiff file for what is now Figure EV3 avoids this problem.

Referee #3:

Review to:

Thermostable ancestors enabled evolutionary diversification of promiscuous chemical defence enzymes

Summary

The authors studied the evolution of CYP2, a p450 multifunctional enzyme family involved in chemical detoxification in vertebrates. The enzymes are multifunctional and known to have evolved through a significant expansion 500 million years ago when the Earth was no longer hot. The authors reconstructed 12 the ancestral states of the protein family. They measured their thermostability and catalytic activity against a battery of substrates. Then, they analyze the structure to understand how thermostability changes over time and how it relates to multifunctionality. The experimental designs aim to prove the hypothesis that multifunctional enzymes are kept thermophilic (robust) to allow their diversification.

The T_m of the 12 ancestors, plus extant enzymes, shows that the young ancestors are less thermostable than the old ancestors. This incremental reduction in T_m was then analyzed in terms of the enzymes' hydrophobic/hydrophilic interactions. They find that the highest thermostability is linked to better fold packing. The authors measure the enzyme's capacity to modify five different substrates and bind 48 chemically diverse ligands. They found that binding promiscuity and the age of the protein do not correlate. However, the most thermostable ancestor birthed the most functionally diverse clades.

Novelty and significance

The research is novel in that it adds to the discussion of the protein community related to "why ancestral proteins are thermostable, even when they are from times when the Earth was not as hot".

The fact that the authors show that is likely not a consensus approach but a lack of pressure to lose the thermostability of older proteins is of value.

Comments

1:

Ancestral sequence reconstruction (ASR) generates more thermostability than the extant

enzymes from where they are inferred. In cases where the ancestors existed on Earth in times when the temperature was indeed high, this makes sense. However, the ASR of enzymes that evolved more recently, when the Earth was no longer hot, also leads to thermostable ancestral problems.

In the introduction, the authors mention two possible explanations: (1) The evolution from thermostable ancestors does not always lead to mesophilic enzymes. Drift can keep them thermostable. (2) Most enzyme families studied by ASR are multifunctional. Hence, they are supposed to stay robust (as in a proxy to thermophilic?) to allow for functional diversification.

Here, the authors should add a third explanation. While constructing ancestral sequences, there could be a bias towards consensus -which leads to thermophilic proteins. They only mention it later in the discussion.

This third hypothesis has now been introduced on p.5 at lines 82-86 with the following text:

“Maximum likelihood methods for ASR have been proposed to modestly overestimate thermostability (Williams, et al. 2006). However the increases seen (up to ~ 30 °C rather than the few degrees predicted by simulations) and the marked difference in sequence between ancestral and consensus inferences argue against these increases being wholly artefactual. The increased thermostability in ancestors has been linked to thermophily...”

2:

Towards the end of the introduction, the authors mention that thermophilic enzymes are supposed to be less multifunctional, as they should be less flexible. This contradicts the hypothesis they want to test, as thermostability should be "penalized" through the evolution of multifunctional enzymes. Perhaps a better explanation here is needed.

The sentence to which the reviewer refers is on p. 6, line 129 – p. 7, line 131 of the original submission:

“In addition, since thermostability may be associated with reduced flexibility in a protein fold, we assessed the relationship between thermostability and substrate promiscuity in this enzyme family.”

The reviewer has nicely summarized the contradiction here that we wanted to address. To clarify the text, we have rephrased as follows (p. 7, lines 133-137):

“In addition, thermostability is commonly assumed to be associated with rigidity, i.e., reduced flexibility in a protein fold. However, rigidity would be evolutionarily disadvantageous in proteins that have to exhibit substrate promiscuity to be functionally relevant. Therefore, we assessed the

relationship between thermostability and substrate promiscuity in this enzyme family.”

3:

Figures 4 and 5 could be merged into 1. Treating all substrates equally regardless of whether they are endogenous or not. Also, some graphs are empty, or the activity can't be easily observed. When the enzymes are inactive, maybe don't add a graph, and for those with very little activity, try to change the Y-axis scale.

In these figures, we aimed to reflect how activity towards structurally similar substrates has changed across the evolutionary tree. All activities are shown relative to the most active ancestor and the scales are already consistently broken to emphasise low levels of activity < 10% maximum, yet in such a way as to not take away from the impression provided by the relative comparison.

We believe it is useful to evaluate the endogenous steroid substrate, testosterone, separately to the xenobiotic probe substrates, since only activity towards the former is relevant in the natural evolution of these enzymes. However, we have consolidated the separate panels of each figure into one to minimize space.

4:

A concluding remark at the end of the results section titled: "Catalytic activity of ancestral forms towards marker substrates" is missing. This was done for all other sections in the results, and I think it helped throughout the reading.

The following comment has been added at the end of this section (p. 21, lines 468-470):

“Collectively, the patterns of activity of ancestors were similar to those seen for the extant forms, i.e., ancestors showed discrete but overlapping ranges of activity towards xenobiotic substrates.”

5:

Discussion, lines 481 - 495: The authors mention that compared to the oldest ancestor, CYP2ABGSFTCEH, newer proteins are more thermostable by 10 degrees. They assume drift increased thermostability. I wonder how significant these 10 degrees are? Maybe some statistical analysis is needed?

A comment about the statistical significance of these differences has been added to the legend of Figure 2 and Appendix Table S3 contains additional details on statistical analysis. The original data and analysis have been deposited, including a matrix showing the statistical significance of all ancestors relative to each other and the extants relative to all their cognate ancestors (Filename: Thermostability & expression data from Supp Table 3 with stats.xlsx). CYP2CEH was significantly more thermostable than CYP2ABGSFTCEH ($p = 0.0151$, two-tailed, heteroscedastic Student's t-test and $p = 0.0076$, one-tailed, heteroscedastic Student's t-test). CYP2CEH was not significantly more thermostable than CYP2ABGSFTCEH ($p = 0.0749$) when analysed using a two-tailed, heteroscedastic Student's t-test but reached significance

at the $p < 0.05$ level when a one-tailed test was used ($p = 0.0374$, one-tailed, heteroscedastic Student's t-test).

6: Discussion, lines 495-500: Why is the lineal decrease in thermostability (Figure 2) proof that thermostability was lost by drift? It was not clear to me.

The comment to which the reviewer refers is on p. 25, lines 570-572 (previously, p 23, lines 502-504):

“These trends are consistent with thermostability being an evolutionary relic from enzymes that existed in very hot environments in primordial times, that was lost through genetic drift in the absence of selection pressure to maintain it.”

We do not claim that Figure 2 proves thermostability was lost by drift. We only say this trend is consistent with this hypothesis. (It is usually impossible to prove hypotheses related to historical evolution.) This hypothesis is also supported (not proven) by the observation that stability is more likely to be lost than gained by evolutionary diversification (Tokuriki, et al. 2008).

The following sentence has been added for clarification on p. 25 lines 572-575:

“Since most mutations are deleterious (Tokuriki, et al. 2008), genetic drift would be expected to lead to a loss of thermostability with accumulation of mutations, until a level was reached where enzyme function began to be impaired.”

7:

Discussion, lines 511-518:

This part is not clear to me. Do the authors mean that organisms today growing in hot environments (thermophiles) have a proteome with a T_m 10-20 degrees Celsius higher than mesophilic? How is that related to the following statement?

The comment was (p. 23, lines 518-525 in the original submission):

“However it is intriguing that an analysis of the thermostability of the proteome of organisms exhibiting different optimal growth temperatures (OGT) showed average protein melting temperatures to be $\sim 10-20$ °C higher than the OGT (Jarzab, et al. 2020). Thus, melting temperatures of 50-60 °C might be expected for proteins in organisms surviving such hothouse environments. While we cannot assign precise ages to any of the hypothetical ancestral proteins resurrected here, all the ancestors which gave rise to more than one subfamily showed $^{60}T_{50}$ values over ~ 59 °C, consistent with this expectation.”

We have rephrased this comment as follows to clarify (p. 26, line 606 – p. 27, line 616):

“Bearing in mind these caveats, it is notable that the average melting temperature of the proteome was ~ 10-20 °C higher than the respective optimal growth temperature (OGT) across a range of organisms exhibiting different OGTs (Jarzab, et al. 2020). The distribution of melting temperatures ranged ~ 10-20 °C above and below the average for the three vertebrates analysed. Thus, average melting temperatures of 50-60 °C would not be unreasonable for proteins in organisms surviving hothouse environments of ~ 30-40 °C and the range could extend up to 80 °C. We cannot assign precise ages to any of the hypothetical ancestral proteins resurrected here, but the deeper ancestors (i.e., CYP2ABGSFTCEH, CYP2ABGSFT, CYP2ABGS, CYP2CEH, CYP2CE and CYP2BS, that could have existed closer to times of hothouse temperatures than the single subfamily ancestors) showed $^{60}T_{50}$ values from ~ 59 – 83 °C, placing them towards the top of, or above, the expected range.”

8:

From the results section, it was hard to visualize the trend the authors refer to in the discussion (lines 571-572), namely, that the ancestors show broader substrate specificity. It seems that they are multifunctional, but also some extant enzymes. So, I'm not sure we can visualize the reduction of multifunctionality between ancestral and extant enzymes with this data.

We had written (p. 25, line 571 – p.26, line 580):

“In general, the extant forms showed greater turnover of the substrates assessed; however, all ancestors showed comparable levels of activity to their extant descendants. When considering just the ancestral forms, the highest turnover for each group of structurally related substrates was generally achieved with one or more younger ancestors, i.e., CYP2A for testosterone and coumarin, CYP2B/ CYP2S for alkoxyresorufins and CYP2E/ CYP2F for luciferin derivatives. Overall, the older ancestors showed low to moderate activity towards multiple substrates, indicating a significant degree of promiscuity.”

The comparison at lines 573-580 was between older and younger ancestors' activity towards xenobiotic substrates, not with extant forms. Also, we had not claimed there was any reduction in multifunctionality in extant forms compared to ancestral. However, we realise that it is difficult to compare across multiple figures in the Appendix information and that the oldest ancestor had the best activity among the ancestors towards two luminogenic substrates. We have simplified this section as follows, to draw out the key point that the ancestors were promiscuous and able to monooxygenate the same substrates as the extant forms (p. 30, lines 689-694):

“All ancestors demonstrated monooxygenation activity towards typical xenobiotic CYP2 substrates using the extant human NADPH-cytochrome P450 reductase as a redox partner. In general, the extant forms showed greater turnover of the substrates assessed; however, the activity shown by the ancestors was in the same range as to their extant descendants overall. Overall, both the older and younger ancestors showed a significant degree of promiscuity.”

We have also removed the following sentences from the results to simplify this section (p. 19, lines 417-422 of the original submission):

“Older ancestors that showed moderate activity towards testosterone such as CYP2CE, CYP2CEH and CYP2ABGSFTCEH also showed moderate activity towards the luciferin derivatives. Ancestors along the CYP2ABGS evolutionary line generally showed the lowest activity towards luciferin substrates, however, these forms i.e. CYP2B, CYP2S and CYP2BS were amongst the most active towards the alkoxyresorufin compounds.”

We have also removed the comment in the abstract about catalytic specificity for simplicity.

9:

Discussion, lines 583-593.

The authors now contradict themselves in the data analysis. Before (point 7), it seems that the ancestral proteins were more promiscuous, but now they write that the ability to bind and monooxygenase compounds is similar between ancestral and extant proteins. But the catalytic rate was lower. A better explanation is needed.

We trust that the simplification of the text in the previous paragraph resolves this contradiction: Both the comparison of activity towards xenobiotic substrates and the inhibition study suggests both extant and ancestral forms are active towards a wide range of structures but that overall, the highest activity towards any given xenobiotic substrate was seen with an extant form.

Additional corrections:

An additional author, Mr Anthony Bengochea, has been added for his role in performing some of the analyses involved in responding to the reviewers' comments.

During the revision process, it was noted that the positions of the CYP2CE, CYP2A, CYP2S and CYP2F nodes were slightly displaced on the tree. These have been corrected now.

A typographical error was detected in the SD for the amplitude of the ticlopidine binding spectrum for CYP2A in Appendix Table S6 and in the node number for the CYP2F ancestor in Appendix Table S3.

Some minor rounding errors in Appendix Table S3 have been corrected.

At the bottom of p. 13, line 273 of the original submission, the extra ‘with’ has been removed (now p. 14 line 291).

A reference has been added to the source of the pCW/1A2/hCPR vector used for cloning.

On p. 15, lines 310-314 and 327-328 of the original submission, reference to amphibian sequences has been removed following clarification of the subfamily designations of these sequences, and the term ‘sauropsid’ has been used to cover reptiles and birds to be more concise (now p. 15, lines 327-330 and p. 16, lines 342-345).

“Indole” has been corrected to “Indigo” in the title of Appendix Figure S7 (formerly Supplementary Figure 8).

On p.30 line 708 (p. 26, line 594 of the original submission), “chemical” has been corrected to “chemicals” and “the likely absorption” has been corrected to “its likely absorption” at line 724 on p. 31.

What were formerly Supplementary figures 1, 10 and 11, along with the expanded view of the phylogenetic tree showing bootstrap values, have been selected for display as extended view figures. The numbering of these and the remaining appendix figures has been adjusted accordingly throughout the text with the extended view figure legends placed at the end of the main text file.

Finally, an acknowledgment has been added to Prof. David R. Nelson and the acknowledgment to Anthony Bengochea has been removed as he is now an author.

References

Abadi S, Azouri D, Pupko T, Mayrose I. 2019. Model selection may not be a mandatory step for phylogeny reconstruction. *Nature Communications* 10:934.

Besenmatter W, Kast P, Hilvert D. 2007. Relative tolerance of mesostable and thermostable protein homologs to extensive mutation. *Proteins: Structure, Function, and Bioinformatics* 66:500-506.

Bloom JD, Labthavikul ST, Otey CR, Arnold FH. 2006. Protein stability promotes evolvability. *Proceedings of the National Academy of Sciences of the United States of America* 103:5869-5874.

Bloom JD, Lu Z, Chen D, Raval A, Venturelli OS, Arnold FH. 2007. Evolution favors protein mutational robustness in sufficiently large populations. *Bmc Biology* 5.

De Vleeschouwer D, Percival LME, Wichern NMA, Batenburg SJ. 2024. Pre-Cenozoic cyclostratigraphy and palaeoclimate responses to astronomical forcing. *Nature Reviews Earth & Environment* 5:59-74.

Foley G, Mora A, Ross CM, Bottoms S, Sützl L, Lamprecht ML, Zaugg J, Essebier A, Balderson B, Newell R, et al. 2022. Engineering indel and substitution variants of diverse and ancient enzymes using Graphical Representation of Ancestral Sequence Predictions (GRASP). *Plos Computational Biology* 18:e1010633.

Gaedigk A, Casey ST, Whirl-Carrillo M, Miller NA, Klein TE. 2021. Pharmacogene Variation Consortium: A Global Resource and Repository for Pharmacogene Variation. *Clin Pharmacol Ther* 110:542-545.

Garcia AK, Schopf JW, Yokobori S-i, Akanuma S, Yamagishi A. 2017. Reconstructed ancestral enzymes suggest long-term cooling of Earth's photic zone since the Archean. *Proceedings of the National Academy of Sciences* 114:4619-4624.

Goldstein RA. 2011. The evolution and evolutionary consequences of marginal thermostability in proteins. *Proteins-Structure Function and Bioinformatics* 79:1396-1407.

Grossman EL, Joachimski MM. 2022. Ocean temperatures through the Phanerozoic reassessed. *Scientific Reports* 12:8938.

Gumulya Y, Baek J-M, Wun S-J, Thomson RES, Harris KL, Hunter DJB, Behrendorff JBYH, Kulig J, Zheng S, Wu X, et al. 2018. Engineering highly functional thermostable proteins using ancestral sequence reconstruction. *Nature Catalysis* 1:878-888.

Jarzab A, Kurzawa N, Hopf T, Moerch M, Zecha J, Leijten N, Bian Y, Musiol E, Maschberger M, Stoehr G, et al. 2020. Meltome atlas—thermal proteome stability across the tree of life. *Nature Methods* 17:495-503.

Judd EJ, Tierney JE, Lunt DJ, Montañez IP, Huber BT, Wing SL, Valdes PJ. 2024. A 485-million-year history of Earth's surface temperature. *Science* 385:eadk3705.

Kirischian N, McArthur AG, Jesuthasan C, Krattenmacher B, Wilson JY. 2011. Phylogenetic and functional analysis of the vertebrate cytochrome P450 2 family. *Journal of Molecular Evolution* 72:56-71.

Konno N, Maeno S, Tanizawa Y, Arita M, Endo A, Iwasaki W. 2024. Evolutionary paths toward multi-level convergence of lactic acid bacteria in fructose-rich environments. *Commun Biol* 7:902.

Muñiz-Trejo R, Park Y, Thornton JW. 2025. Robustness of ancestral sequence reconstruction to among-site and among-lineage evolutionary heterogeneity. *Molecular Biology and Evolution* 42:msaf084.

Ngcobo PE, Nkosi BV, Chen W, Nelson DR, Syed K. 2023. Evolution of Cytochrome P450 Enzymes and Their Redox Partners in Archaea. *International Journal of Molecular Sciences*. doi: 10.3390/ijms24044161

Robert F, Chaussidon M. 2006. A palaeotemperature curve for the Precambrian oceans based on silicon isotopes in cherts. *Nature* 443:969-972.

Tokuriki N, Stricher F, Serrano L, Tawfik DS. 2008. How protein stability and new functions trade off. *Plos Computational Biology* 4:e1000002.

Trudeau DL, Kaltenbach M, Tawfik DS. 2016. On the potential origins of the high stability of reconstructed ancestral proteins. *Molecular Biology and Evolution* 33:2633-2641.

Dear Prof. Gillam,

Thank you for submitting your manuscript for consideration by the EMBO Journal.

I am happy to tell you that it is accepted in principle. Yet, according to our regulations I cannot formally accept it before some last few issues are resolved.

Here are the issues raised by our editorial assistance team:

*Keywords: Currently 8 are included. Please limit the number of keywords to 5

*AUTHORS: email bounced for James Beckett - j.beckett@uq.edu.au => Please update the author's profile in our system with the author's current email address.

*AFFILIATIONS: 3 authors are employed by Astra Zeneca; please include this information in the Disclosure and Competing Interests Statement.

*Author Contributions: Please remove them from the manuscript text and make sure to provide the full list by entering the information in our system.

*DisclCIS: please rename according to our standards.

*APPENDIX 1 FILE WITH ToC: Was missing but now uploaded; please add page numbers in the table of contents.

*REAGENT TABLE: Missing. Please complete the reagents table and upload it as a separate file.

*DATA NOT SHOWN: on p. 50 and 51 but if data is indeed accessible in the specified appendix figures, this should be OK. Please check and verify.

*SYNOPSIS IMAGE: Please upload a graphical abstract, sized 550 pixels wide x 300 - 600 pixels high, in jpg or tif(f) format

*SYNOPSIS TEXT: Please provide a summary statement of the manuscript's main findings, together with 3 - 5 bullet points and upload this as a separate document. You can look on several recent issues for examples.

*FIGURE CALLOUTS: Please add citations for panels A,B for Figure 3, as well as for the panels in the EV figures. Please note that Fig EV2 is not within our figure formatting specifications and there will be challenges for the typesetters. Please consider making this an appendix figure instead to avoid any issues going forward.

Additional specific notes:

- Please remove the list of abbreviations and incorporate them into the manuscript text

- Please replace the heading "Main" with "Introduction"

- Please correct the order and headings of the manuscript sections to the following:

Abstract / Keywords / Introduction / Results / Discussion / Methods / Data Availability / Acknowledgements / Disclosure and Competing Interests Statement / References / Figure Legends / Expanded View Figure Legends

Please note that clicking on the Figshare link provided in the manuscript (https://figshare.com/articles/dataset/_/28524209) results in a Page Not Found message. This has to be fixed before acceptance of your manuscript.

- Figure legends:

1. Please note that information related to n is missing in the legends of figure EV2 B. Please correct.

2. Please note that the error bars are not defined in the legends of figure EV2 B. Please correct.

3. Please note that for heatmap present in figure 3A a numbered scale bar is not provided. This needs to be rectified.

We generally allow three months as standard revision time. Yet, in light of the minor nature of these issues I am confident you would be able to submit your revised manuscript much earlier.

Thank you for the opportunity to consider your work for publication. I look forward to your final revision.

Yours sincerely,

Yehu Moran
Academic editor
The EMBO Journal

We realize that it is difficult to revise to a specific deadline. In the interest of protecting the conceptual advance provided by the work, we recommend a revision within 3 months (26th Jan 2026). Please discuss the revision progress ahead of this time with the editor if you require more time to complete the revisions.

The authors addressed the remaining editorial issues.

Dear Prof. Gillam,

I am pleased to inform you that your manuscript has been accepted for publication in the EMBO Journal.

You may qualify for financial assistance for your publication charges - either via a Springer Nature fully open access agreement or an EMBO initiative. Check your eligibility: <https://link.springer.com/journal/44318/how-to-publish-with-us>

Yours sincerely,

Yehu Moran
Academic Editor
The EMBO Journal

Please note that it is The EMBO Journal policy for the transcript of the editorial process (containing referee reports and your response letters) to be published as an online supplement to each paper. If you should prefer removal of any referee-only figures included in the point-by-point response(s), e.g. because they may still be used for future publication or because they have been reproduced from published work by others, please do let us know immediately via response email.

More information is available here: <https://link.springer.com/partners/embo-press/editorial-policies#Peer%20review>